

## Pan-Eurasian Experiment (PEEX):

## Towards a holistic understanding of the feedbacks and interactions in the land - atmosphere - ocean- society continuum in the Northern Eurasian region

Hanna K. Lappalainen[1,2], Veli-Matti Kerminen[1], Tuukka Petäjä[1], Theo Kurten[3], Aleksander Baklanov[4,5], Anatoly Shvidenko[6], Jaana Bäck[7], Timo Vihma[2], Pavel Alekseychik[1], Meinrat O. Andreae[8,] Stephen R. Arnold[9], Mikhail Arshinov[10], Eija Asmi[2], Boris Belan[10], Leonid Bobylev[11], Sergey Chalov[12], Yafang Cheng[13], Natalia Chubarova[12], Gerrit de Leeuw[1,2], Aijun Ding[13], Sergey Dobrolyubov[12], Sergei Dubtsov[14], Egor Dyukarev[15], Nikolai Elansky[16], Kostas.Eleftheriadis[17], Igor Esau[18], Nikolay Filatov[19], Mikhail Flint[20], Congbin Fu[13], Olga Glezer[21], Aleksander Gliko[22], Martin Heimann[23], Albert A. M. Holtslag[24], Urmas Hõrrak[25], Juha Janhunen[26], Sirkku Juhola[27], Leena Järvi[1], Heikki Järvinen[1], Anna Kanukhina[28], Pavel Konstantinov[12], Vladimir Kotlyakov[29], Antti-Jussi Kieloaho[1], Alexander S. Komarov[30], Joni Kujansuu[1], Ilmo Kukkonen[31], Ella-Maria Duplissy[1], Ari Laaksonen[2], Tuomas Laurila[2], Heikki Lihavainen[2], Alexander Lisitzin[20], Alexsander Mahura[5], Alexander Makshtas[32], Evgeny Mareev[33], Stephany Mazon[1], Dmitry Matishov[34,†], Vladimir Melnikov[35,36], Eugene Mikhailov[37], Dmitri Moisseev[1], Robert Nigmatulin[20], Steffen M. Noe[38] Anne Ojala[7], Mari Pihlatie[1], Olga Popovicheva[39], Jukka Pumpanen[40], Tatjana Regerand[19], Irina Repina[16], Aleksei Shcherbinin[27], Vladimir Shevchenko[20], Mikko Sipilä[1], Andrey Skorokhod[16], Dominick V. Spracklen[9], Hang Su[13], Dmitry A. Subetto[19], Junying Sun[41], Arkady Yu. Terzhevik[19], Yuri Timofeyev[37], Yuliya Troitskaya[33], Veli-Pekka Tynkkynen[42], Viacheslav I. Kharuk[43], Nina Zaytseva[22], Jiahua Zhang[44], Yrjö Viisanen[2], Timo Vesala[1], Pertti Hari[7], Hans Christen Hansson[45], Gennady G. Matvienko[10], Nikolai S. Kasimov[12], Huadong Guo[44], Valery Bondur[46], Sergej Zilitinkevich[1,2,12,33] and Markku Kulmala[1]

Department of Physics, University of Helsinki, 00014 Helsinki, Finland
Finnish Meteorological Institute, Research and Development, 00101 Helsinki, Finland
Department of Chemistry, University of Helsinki, 00014 Helsinki, Finland
World Meteorological Organization, 1211 Genève, Switzerland
Danish Meteorological Institute, Research and Development Department, 2100, Copenhagen
International Institute for Applied Systems Analysis, 2361 Laxenburg, Austria
Department of Forest Sciences, University of Helsinki, 00014 Helsinki, Finland
Biogeochemistry and Multiphase Chemistry Departments, Max Planck Institute for Chemistry, 55020 Mainz, Germany
Institute for Climate and Atmospheric Science, School of Earth and Environment, University of Leeds, Leeds, LS2 9JT, UK
Institute of Atmospheric Optics, Russian Academy of Sciences, Tomsk 634021, Russia
Nansen International Environmental and Remote Sensing Center, St. Petersburg, Russia
Lomonosov Moscow State University, Faculty of Geography, Moscow 119899, Russia
Institute for Climate and Global Change Research & School of Atmospheric Sciences, Nanjing University, 210023 Nanjing, China
Institute of Chemical Kinetics & Combustion, Russian Academy of Sciences, 630090 Novosibirsk, Russia
Institute of Monitoring of Climatic & Ecological Systems SB RAS, 634055 Tomsk, Russia
A.M. Obukhov Institute of Atmospheric Physics, Russian Academy of Sciences, Russia
National Centre of Scientific Research "DEMOKRITOS", Greece
Nansen Environmental and Remote Sensing Center/Bjerknes Centre for Climate Research,5006 Bergen, Norway
Northern Water Problems Institute, Karelian Research Center, Russian Academy of Sciences,185003 Petrozavodsk, Russia
P.P. Shirshov , Institute of Oceanology, Russian Academy of Sciences, Russian Academy of Sciences, 117997 Moscow, Russia
Institute of Geography, Russian Academy of Sciences, Moscow, Russia
Department of Earth Sciences of the Russian Academy of Sciences, Russian Academy of Sciences, 119991,  Moscow, Russia
Max-Planck-Institute for Biogeochemistry, 07745 Jena, Germany
Wageningen University, 6708 Wageningen, Nederland
Institute of Physics, University of Tartu, 18 Ülikooli St., 50090 Tartu, Estonia.
University of Helsinki, Department of World Cultures, 00014 Helsinki, Finland
Department of Environmental Sciences, University of Helsinki, 00014 Helsinki, Finland
Russian State Hydrometeorological University, 195196 Saint Petersburg, Russia
Institute of Geography, Russian Academy of Sciences, Moscow, Russia
Institute of Physico-chemical & Biological Problems in Soil Science, Russian Academy of Sciences, 142290 Institutskaya, Russia
University of Helsinki, Geophysics and Astronomy, 00014 Helsinki, Finland
Actic & Antarctic Research Institute, Russian Academy of Sciences, St. Petersburg 199397, Russia
Department of Radiophysics, Nizhny Novgorod State University, Nizhny Novgorod, Russia
Southern Center of Russian Academy of Sciences, Rostov on Don, Russia, [†] deceased, 20.August.2015
Tyumen Scientific Center, Siberian Branch, Russian Academy of Science, Russia
Tyumen State University, 625003 Tyumen, Russia
Saint Petersburg State University, 7/9 Universitetskaya nab., St. Petersburg, 199034 Russia
Institute of Agricultural and Environmental Sciences, Estonian University of Life Sciences, 51014 Tartu, Estonia
Skobeltsyn Institute of Nuclear Physics, Moscow State University, Department Microelectronics, Russia
University of Eastern Finland, Department of Environmental Science, P.O.Box 1627 FI-70211 Kuopio, Finland
Graduate University of Chinese Academy of Sciences, 100049 Beijing, China
Aleksanteri Institute and Department of Social Research, 00014 University of Helsinki, Finland
Sukachev Forest Institute, Russian Academy of Sciences, Krasnoyarsk 660036, Russia
Institute of Remote Sensing and Digital Earth, Chinese Academy of Sciences, Beijing, 100094, China
Environmental Science and Analytical Chemistry, Stockholm University, Sweden
AEROCOSMOS Research Institute for Aerospace Monitoring, 105064, Moscow, Russia
*Correspondence to:* Hanna K. Lappalainen (hanna.k.lappalainen(at)helsinki.fi)
Abstract. The Northern Eurasian regions and Arctic Ocean will very likely undergo substantial
changes during the next decades. The arctic-boreal natural environments play a crucial role in
the global climate via albedo change, carbon sources and sinks, as well as atmospheric aerosol
production from biogenic volatile organic compounds. Furthermore, it is expected that global
trade activities, demographic movement, and use of natural resources will be increasing in the
Arctic regions. There is a need for a novel research approach, which not only identifies and
tackles the relevant multi-disciplinary research questions, but is also able to make a holistic
system analysis of the expected feedbacks. In this paper, we introduce the research agenda of
the Pan-Eurasian Experiment (PEEX), a multi-scale, multi-disciplinary and international
program started in 2012 (https://www.atm.helsinki.fi/peex/). PEEX is setting a research
approach where large-scale research topics are investigated from a system perspective and
which aims to fill the key gaps in our understanding of the feedbacks and interactions between
the land-atmosphere-aquatic-society continuum in the Northern Eurasian region. We introduce
here the state of the art of the key topics in the PEEX research agenda and give the future
prospects of the research, which we see relevant in this context.
Contents

1. Introduction

The global environment is changing rapidly due to anthropogenic influences. As a result, we are already facing several "Grand Challenges" in the 21st century (e.g., Smith 2010; Bony et al., 2015, IPCC; Randers 2012). Two of these challenges, climate change and air quality, are strongly influenced by human activities and their impacts on changing atmospheric composition, more specifically on the concentrations of greenhouse gases, reactive trace gases, and aerosol particles. In the future, the arctic-boreal natural environment will play a crucial role in the global climate via albedo changes, carbon sources and sinks, as well as aerosol production from biogenic volatile organic compounds (Arneth et al., 2010; 2014; Ballantyne et al., 2012; Carslaw et al., 2010; Kulmala et al., 2014).

In order to advance our understanding on interlinked grand challenges further, we need a research approach that helps us to construct a holistic scientific understanding of the feedbacks and interactions within the continuum of land-atmosphere-aquatic-systems and society across different spatial and temporal scales. Therefore, we have established the Pan-Eurasian Experiment (PEEX) program (https://www.atm.helsinki.fi/peex/), which is a multi-disciplinary, multi-scale research initiative focusing on understanding biosphere-ocean-cryosphere-climate-society interactions and feedbacks (Lappalainen et al., 2014, Kulmala et al., 2015). PEEX fills some of the most critical scientific gaps needed for a holistic understanding of the feedback mechanisms characteristic of the Northern Eurasian

geographical domain. Boreal forests and peat lands characterize the vast land areas of Northern
Eurasia, with a major part of them situated in the Russian territory. In addition to natural
environments, the PEEX research program is also interested in different human-influenced
environments: from urban to countryside, from megacities to non-populated remote areas,
from areas of dispersed settlements and sparsely-built environments to heavily-industrialized
regions. Thus, the research approach covers the Arctic and boreal regions situated in Northern
Eurasia, and also the marine environments of the Arctic Ocean. PEEX operates in an
integrative way using tools from natural and social sciences such as in-situ and satellite
observations, laboratory experiments, multi-scale models, and statistical data analyses,
together with socio-economic analyses. The PEEX research agenda covers spatial scales from
regional to global and temporal scales and  from seconds to decades (Kulmala et al., 2011b).
The scientific results will be used for developing new climate scenarios on global and regional
scales, for constructing reliable early-warning systems, and for the mitigation and adaptation
planning of the Northern societies in the most efficient way. PEEX aims to contribute to
climate policy concerning topics important to the Northern Eurasian environment,, helping
societies in building a sustainable future.
2.System perspective approach

Earth (System) Sciences (ESS) has emerged as one of the most rapidly developing

scientific fields. The recent growth of ESS has been facilitated by the importance of
understanding the fundamental scientific processes of climate change and air quality, as well
as the increasing societal impact of this research area. The development has mainly taken place
among natural sciences, while the collaboration between natural and social sciences to tackle
climate change issues has started to emerge relatively slowly. A multi- and cross-disciplinary
approach is thus needed to advance the solution-oriented understanding of grand challenges
and to apply new knowledge for reliable climate scenario development, mitigation and
adaptation, as well as early warning system development. In addition to enhanced
collaboration between different branches of science, there is a need for a next generation of
multidisciplinary scientists able to connect the scientific issues with an understanding of the
societal dimensions related to the grand challenges.
Climate change can be considered as the main driving force for system changes and their
feedback dynamics, especially in the Arctic-boreal regions. It has already been estimated that
the future warming in Northern high latitudes regions will be, on average, larger than that
experienced at lower latitudes (IPCC, 2013, 2014). The climate change driven processes taking
place in the Arctic provide a good example on how important it is to quantify feedback
dynamics and at the same time study the specific research topics from the land-atmosphere-
hydrosphere-cryosphere-societal system perspective. For example, the surface radiation
balance regulates the melting and freezing of the pack ice, which in turn is a key climate
regulator. Model simulations of Arctic clouds are particularly deficient, impeding correctly
simulated radiative fluxes, which are vital for the estimation of the snow/ice-albedo feedback
(Vavrus et al., 2009). Important, yet poorly-quantified players in the Arctic atmospheric
system and climate change are the short-lived climate forcers (SLCF), such as black carbon
and ozone. The climatic impacts of SLCFs are tightly connected with cryospheric changes of
the land system, and associated with human activities. Models display diverse and often poor
skill in simulating SLCF abundances both at the surface and vertically through the troposphere
at high latitudes (Eckhardt et al., 2015; Emmons et al., 2015; Monks et al., 2015).
PEEX is setting a research approach where the large-scale research questions are studied
from a system perspective, and which is also filling the key gaps in our understanding of the
feedbacks and interactions between the land, atmosphere, aquatic, and societal systems in the
Northern Eurasian region (Kulmala et al., 2015). We have structured the research agenda so
that we have highlighted three thematic research areas per system (Fig. 1). The identification
of these key thematic research areas has been based on a bottom-up approach by researchers
coming from Europe, Russia, and China, participating PEEX meetings and conferences
starting from 2012. These researchers first introduced a wide spectrum of specific research
topics relevant to the Northern Eurasian region, which were then evaluated and classified. This
bottom up process led to the so-called "system-based" structure with altogether twelve
thematic research areas. This approach will piece by piece lead into a holistic system
understanding, quantifying the dominant feedbacks and interactions between the systems, and
providing an understanding of the dynamics of Arctic-boreal biogeochemical cycles (e.g.,
water, carbon, nitrogen, phosphorus, sulfur). In our approach, climate change is the key driver
in the dynamics of the land, atmosphere, aquatic and societal systems (Kulmala et al., 2015).
The large-scale thematic areas of each system and many of the research highlight topics
introduced by the PEEX research agenda are fundamentally related to climate-change-driven
shifting GHG and SLCF formation processes and their primary and secondary feedbacks
between socioeconomic and biogeochemical systems. When studying the Arctic-boreal
feedback loops in a wider context, the PEEX agenda addresses China as the most crucial source
area of atmospheric pollution, having a significant impact on the chemical composition of
the atmosphere over Northern Eurasia (Monks et al, 2015). One must keep in mind that solving
air quality – climate interactions is also the key to practical solutions on local air quality
problems in China.
In this paper, we introduce the state of the art of the selected thematic research areas and
summarize the future research needs at large scale. This introduction serves a White Paper of
the PEEX research community. The thematic research areas relevant to the Land System are
related to "changing ecosystem processes" (2.1.1), "ecosystem structural changes and
resilience" (2.1.2), and "risk areas of permafrost thawing" (2.3.1). In the Land System research
agenda, we address the following key issues: changing boreal forests biomass, Arctic greening,

and permafrost processes. The main research areas of the Atmospheric System research are "the specific characterization of the atmospheric composition and chemistry" (2.2.1), "urban air quality" (2.3.2.), and "the atmospheric circulation and weather" (2.2.3). In terms of atmospheric systems, we address oxidants, trace gases, greenhouse gases, and aerosols as atmospheric key components. We highlight that future advances in predicting urban air quality and improving weather forecasting are strongly based the the atmospheric boundary layer dynamics research (Holtslag et al., 2013).

The thematic research areas relevant to the Aquatic System are "the Arctic Ocean in the climate system" (2.3.1), "the Arctic maritime ecosystems" (2.3.2), and "the lakes, wetland and rivers systems" (2.3.3). Under these research areas, we focus on topics like Arctic sea ice changes, marine gross primary production, and Arctic pelagic food webs under environmental changes. Lakes and large-scale river systems have multiple roles and aspects of the physical environments, starting from water chemistry and algal blooms, and ending up with carbon and methane dynamics.

The thematic areas of the Societal System have a number of dimensions, but in the first phase the primary interest lies on studying the consequences of "the increasing use of natural resources" (2.4.1), on the growing number of "natural hazards" (2.4.2), and on "the social transformations" (2.4.3) in the Northern Eurasian region. We see topics like the future Siberian forest area together with fuel balance, forest fires effecting the carbon and nitrogen balance, and societal dimensions related to infrastructure degradation as the most important future research areas. In Section 3, we investigate the connections and interlinks between those four systems.

2.1. Land system – state of the art and future research needs

2.1.1. Changing land ecosystem processes

In the future, many Arctic-boreal processes will respond sensitively to climate change,
affecting ecosystem productivity and functions. These changes may lead to unprecedented
consequences, e.g., in the magnitude of the ecosystem carbon sinks, production of aerosol
precursor gases, and surface albedo. We need first to develop methods for identifying the land
regions and processes that are especially sensitive to climate change. Only after that are we
able to analyze their responses.
Boreal forests are one of the largest terrestrial biomes, and account for around one third
of the Earth's forested area (Global Forest Watch, 2002; http://www.globalforestwatch.org/).
Nearly 70 % of all boreal forests are located in the Siberian region. The forest biomass, soils,
and peatlands in the boreal forest zone together constitute one of the world's largest carbon
reservoirs (Bolin et al., 2000; Kasischke, 2000; Schepaschenko et al., 2013). Due to their large
forest surface areas and huge stocks of carbon (~320 gigatonnes of carbon; GtC), the boreal
and Arctic ecosystems are significant players in the global carbon budget. Furthermore,
permafrost, a dominant feature of Siberian landscapes, stores around 1700 GtC (Tarnocai et
al., 2009). Boreal forests form the main vegetation zone in the catchment areas of large river
systems, so they are an important part of the global water-energy-carbon feedbacks.
The forest biomass forms a climate feedback via the anticipated changes in nutrient
availability and temperatures, affecting carbon sequestered into both the aboveground biomass
and soil compartment. The Siberian forests are currently assumed to be a carbon sink, although
with a large uncertainty range of 0-1 PgC $yr^{-1}$ (Gurney et al., 2002). However, these
ecosystems are vulnerable to global climate change in many ways, and the effects on
ecosystem properties and functioning are complicated. While higher ambient $CO_2$
concentrations and longer growing seasons may increase plant growth and productivity, as
well as the storage of carbon to soil organic matter (e.g., Ciais et al., 2005, Menzel et al., 2006),
warming affects respiration and ecosystem water relations in the opposite way (Bauerle et al.,
2012; Parmentier et al., 2011). Expected acceleration of fire regimes might also substantially
impact the carbon balance in Arctic and boreal regions (Shvidenko and Schepaschenko, 2013).
One example of the potentially large feedbacks is the critical role that permafrost plays
in supporting the larch forest ecotone in northern Siberia. The boreal forests in the high
latitudes of Siberia are a vast, rather homogenous ecosystem dominated by larch. The total
area of larch forests is around 260 million ha, or almost one-third of all forests in Russia. Larch
forests survive in the semi-arid climate because of the unique symbiotic relationship they have
with permafrost. The permafrost provides enough water to support larch domination, and the
larch in turn blocks radiation, protecting the permafrost from intensive thawing during the
summer season. The anticipated thawing of permafrost could decouple this relationship, and
may cause a strong positive feedback, intensifying the warming substantially.
The ambient temperature, radiation intensity, vegetation type, and foliar area are the
main constraints for the emission of biogenic volatile organic compounds (BVOCs)
(Laothawornkitkul et al., 2009). This makes BVOC emissions sensitive to both climate and
land use changes, via, e.g., increased ecosystem productivity or the expansion of forests into
tundra regions. Although the inhibitory effect of $CO_2$ on the process level may be important,
Arctic greening may strongly enhance the production of BVOCs in northern ecosystems
(Arneth et al., 2007; Sun et al., 2013). Open tundra may also act as a significant source for
BVOCs, especially if the snow cover period changes (Aaltonen et al., 2012; Faubert et al.,
2012). This would lead to negative climate feedbacks involving either aerosol-cloud or
aerosol-carbon cycle interactions (Kulmala et al., 2013; 2014; Paasonen et al., 2013). Linear
trends in the annual maximum Normalized Difference Vegetation Index (NDVI) over 15 years
in the Northern areas of the Yamalo-Nenets, Okrug region in Russia, provide supporting
evidence of the increasing biological activity and greening,  and the potential for enhanced
BVOC emissions (Fig.2).

In summary, even small proportional changes in ecosystem carbon uptake can switch

terrestrial ecosystems from a net carbon sink to a carbon source, with consequent impacts on
atmospheric $CO_2$ concentrations and global temperatures (e.g., Bala et al., 2013; Bodman et
al., 2013, Mukhortova et al., 2015). This process has already been observed, particularly in
disturbed forests of Northern Asia (Shvidenko and Schepaschenko 2014). Currently, we do
not fully understand all the factors influencing carbon storage, or the links between
biogeochemical cycles of carbon, water, and nutrients in a changing climate. However, the
changes in these processes may be large, and their impacts may either amplify or decrease
climate change, especially in the high northern latitudes.
2.1.2  Ecosystem structural changes and resilience
The ecosystem structural changes are tightly connected to adaptation needs, and to the

development of effective mitigation and adaptation strategies. Predictions concerning the
shifting of vegetation zones are important for estimating the impacts of the region on future
global GHG, BVOC, and aerosol budgets. Furthermore, natural and anthropogenic stresses,
such as land use changes and biotic and abiotic disturbances, are shaping ecosystems in the
Arctic and boreal regions and have many important feedbacks to climate (see e.g., the review
by Gauthier et al., 2015). In a warmer climate, northern ecosystems may become susceptible
to insect outbreaks, drought, devastating forest fires, and other natural disasters. In addition,
human impacts may cause sudden or gradual changes in ecosystem functioning. The
ecosystem resilience is dependent on both the rate and magnitude of these changes. Recent
studies have concluded that current estimates very likely overestimate the resilience of global
forests and particularly boreal forests (Allen et al., 2015). In some cases, the changes may lead
to system imbalance and to the crossing of a tipping point, after which the effects are
irreversible. One of the most relevant research topics for the land system are to determine the
structural changes and tipping points of the ecosystem changes in the Northern Pan Eurasian
region.

Part of the expected ecosystem structural changes is related to the lengthening of the

growing season that is taking place the Arctic-boreal regions due to climate change. This
phenomenon, called "Arctic Greening", is due to increased plant biomass growth and
advancing tree lines, turning previously open tundra into shrubland or forest (Myneni et al.,
1997; Xu et al., 2013). However, "browning" as a proxy of decreased productivity has also
been observed during recent decades in many boreal regions (Lloyd and Bunn 2007), including
vast territories of Central Siberia, together with a general downward trend in basal area
increment after the mid-20th century (Berner et al., 2013) and the overall decline in green from
2011 to 2014 in Arctic regions (Phoenix and Bjerke 2016). Current predictions on the extent
and magnitude of these processes vary significantly (Tchebakova et al., 2009; Hickler et al.,
2012; Shvidenko et al., 2013). It has been estimated that the northward shift of bioclimatic
zones in Siberia will be as large as 600 km by the end of this century (Tchebakova et al., 2009).
By taking into account that the natural migration rate of boreal tree species cannot exceed 200-
500 m per year, such a forecast implies major vegetation changes in huge areas. In addition,
we need to have a deeper understanding of the future role of the browning process and to re-
analyze the previous model predictions of arctic greening: to what extent are they wrong, and
why (Phoenix and Bjerke 2016)? This has important biophysical consequences and climatic
feedbacks. Changes in vegetation cover can, e.g., lead to albedo changes and therefore higher
net absorption of radiation in regions covered by forests compared to open vegetation (Jeong
et al., 2011). This modifies the local heat and vapor fluxes, and affects boundary layer
conditions as well as both local and larger-scale climate (Sellers et al., 1997).

Northern peatlands contain a significant part of the global soil organic matter reservoirs

(45% of the world's soil carbon; Post et al., 1982), and comprise one of the world's largest
GHG sources (in particular CH$_4$) (IPCC 2013). The hydrological conditions are a major factor
in determining the functioning of peatlands as carbon source or sink, and the carbon balance
of the vast northern peatlands is extremely sensitive to human influence, be it through either
management or climate change. For example, thawing of permafrost peatlands in tundra
regions might change tundra ecosystems from a stable state into a dynamically changing and
alternating land-water mosaic, with dramatic impacts on their GHG production (Heikkinen et
al., 2004; Repo et al., 2009). Today, peatland management activities range from drainage and
peat harvesting to establishing crop plantations and forests. A complete understanding of the
climatic effects of peatland management remains a challenging question (Maljanen et al.,

2010).

Northern ecosystems are frequently suffering from increased stresses and deterioration.
There is seldom a single and clear cause for forest dieback, but rather the ecosystems are
suffering from multiple stresses simultaneously (e.g., Kurz et al., 2008 a,b; Allen et al., 2010).
This implies that a single stress factor may not be very dramatic for the resilience of the system,
but when occurring simultaneously in combination with others, the system may cross a
threshold (i.e., tipping point), and this may have dramatic consequences. Such perturbations
and disturbances can include long-term pollutant exposures, but also stochastic events such as
fires, flooding, windstorms, or insect population outbreaks, and human activities such as
deforestation or the introduction of exotic plant or animal species. Disturbances of sufficient
magnitude or duration can profoundly affect an ecosystem, and may force an ecosystem to
reach a threshold beyond which a different regime of processes and structures predominates.
Climate warming, precipitation changes during growth periods, and permafrost changes will
substantially increase water stress, and consequently increase the risk of mortality for trees.
This process has already clearly intensified over the entire circumpolar boreal belt (Allen et

al., 2010). As a consequence, ecosystems may turn into carbon sources rather than sinks (Parmentier et al., 2011).

In the future, boreal forest diebacks may occur due to mass infections of invasive pathogens or herbivores, such as the autumnal moth (*Epirrita autumnata*) or mountain bark beetle (*Dendroctonus ponderosae*), that have previously been climatically controlled by harsh winter conditions. The growth and life cycles of herbivores or their habitat conditions may change in such a way that the outbreak frequencies and intensities of previously relatively harmless herbivore populations increase (Hunter et al., 2014). At the same time as climate is changing, boreal vegetation is also exposed to increased anthropogenic influences by pollutant deposition and land use changes (Dentener et al., 2006; Bobbink et al., 2010; Savva and Berninger, 2010). Large industrial complexes may lead to local forest diebacks, as has been observed in the Kola region (e.g., Nöjd and Kauppi, 1995; Tikkanen, 1995; Kukkola et al., 1997) and in some regions of Siberia (Baklanov et al., 2013). Societal transformations may lead to abandonment of agricultural land or deterioration of previously managed forests.

### 2.1.3. Risk areas of permafrost thawing

The major part of the Northern Eurasian geographical region is covered by continuous permafrost. The fate of permafrost soils in high latitudes is important for global climate with regard to all greenhouse gases. Thawing of permafrost will also substantially alter the hydrological regimes, particularly in Northern Asia, which will lead to increasing water stress in forests and explosive enlargement of fire extent and severity as well as post fire successions (Shvidenko et al., 2013). These scenarios underline the urgent need for systematic permafrost monitoring, together with GHG measurements in various ecosystems. The treatment of permafrost conditions in climate models is still not fully developed (Bala et al., 2013). The major question is, how fast will the permafrost thaw proceed and how will it affect ecosystem

processes and ecosystem-atmosphere feedbacks, including hydrology and greenhouse gas
cycling.

Understanding of the feedbacks between carbon and water cycling, ecosystem

functioning, and atmospheric composition related to permafrost thawing is one of the
important topics of the land system study (Heimann and Reichstein, 2008; Schuur et al., 2009;
Arneth et al., 2010). In high-latitude ecosystems with large, immobile carbon pools in peat and
soil, the future net $CO_2$ and $CH_4$ exchange will depend on the extent of near-surface permafrost
thawing, local thermal and hydrological regimes, and interactions with the nitrogen cycle
(Tarnocai et al., 2009). The extra heat produced during microbial decomposition could
accelerate the rate of change in active-layer depth, potentially triggering a sudden and rapid
loss of carbon stored in carbon-rich Siberian pleistocene loess (yedoma) soils (Khvorostyanov
et al., 2008).

The connection between the climate and the thermal conditions in the subsurface layers

(soil and bedrock) is an important aspect. The warming of the atmosphere will inevitably result
in the warming of the permafrost layer, and is easily observed in deep borehole temperature
data. However, the changes depend on the soil and rock type as well as on the pore-filling
fluids. As long as the pore-fill is still ice, the climatic changes are reflected mainly in the
thickness of the active layer, and in slow diffusive temperature changes of the permafrost layer
itself. In areas where the ground is dominated by low ground temperatures and thick layers of
porous soil types (e.g., sand, silt, peat), the latent heat of the pore filling ice will efficiently
'buffer' and retard the final thawing. This is one of the reasons why relatively old permafrost
exists at shallow depths in high-porosity soils. On the other hand, quite different conditions
prevail in low-porosity areas, e.g., in crystalline rock areas.

The permafrost dynamics affect methane fluxes in many ways. Hot spots such as mud

ponds emitting large amounts of $CH_4$ may form when permafrost mires thaw. In contrast, lakes

have occasionally disappeared as a result of the intensification of soil water percolation (Smith et al., 2005). The rapid loss of summer ice, together with increasing temperature and melting ice deposits, results in coastal erosion, physical destruction of surface in hilly areas, activation of old carbon and elevated $CO_2$ and $CH_4$ emissions from sea bottom sediments (Vonk et al., 2012). High methane emissions have been observed from the East Siberian Arctic self (Shakhova et al., 2010).

2.2. Atmospheric system - state of the art and future research needs

2.2.1. Atmospheric composition and chemistry

Atmospheric composition plays a central role in the Northern Eurasian climate system. In addition to greenhouse gases and their biogeochemical cycling discussed in more detail in section 3.2, key compounds in this regard are ozone and other oxidants, carbon monoxide, numerous organic compounds, as well as different types of aerosols and their precursors ($SO_2$ will be discussed in chapter 3). At the moment, there is a serious gap in our knowledge on tropospheric composition and chemistry over Russia and China, with particularly few observation programs being active over Siberia (Crutzen et al., 1998; Ramonet et al., 2002; Paris et al., 2008; Kozlova et al., 2008; Uttal et al., 2015, Paris et al., 2010 a,b;  Sasakawa et al., 2010;  Chi et al., 2013; Saeki et al., 2013; Ding et al., 2013a, 2013b; Berchet et al., 2015; Heimann et al., 2014).

There is thus an urgent need for harmonized, coordinated and comprehensive greenhouse gas, trace gas and aerosol in-situ observations over Northern Eurasia and China (long-term transport aspect) comparable to European and circumpolar data observations. In Fig. 3 we illustrate the geographical coverage of the ground stations that will be part of the coordinated, coherent, and hierarchic observation network in the Northern Eurasian region and in China.

### 2.2.1.1. Main pollutants

Little is known about whether and how the regional ozone budget in northern Pan-Eurasia differs from that in the rest of the northern hemisphere (Ding et al., 2008; Berchet et al., 2013). Arctic tropospheric ozone is significantly influenced by long-range import of ozone and precursors from mid-latitude sources, as well as by boreal wildfires (Ding et al., 2009; Wespes et al., 2012; Paris et al., 2010b; Vivchar et al., 2009). The role of biomass burning emissions in the ozone budget in high latitudes remains controversial. While most studies suggest significant ozone production in boreal smoke plumes (e.g., Paris et al., 2010b; Parrington et al., 2013; Jolleys et al., 2015), some observations from individual plumes suggest that $O_3$ production in boreal wildfire plumes may be weaker, or even turn into net destruction, compared to fire plumes at lower latitudes (Liang et., 2011; Jaffe and Wigder, 2012). Recent modeling work has suggested that boreal fires produce a substantial large-scale enhancement in summertime ozone at high latitudes, which appears to be highly sensitive to differences in partitioning of reactive nitrogen among models (Arnold et al., 2015). The boreal biosphere, on the other hand, provides a large sink for tropospheric ozone (Paris et al, 2010b; Parrington et al., 2013). Given their importance for air quality and global greenhouse gas budget, more atmospheric measurements of $O_3$, its precursors and other pollutants over Siberia are needed (see Elansky, 2012). This is particularly the case in light of increasing local Arctic sources of ozone precursors (NOx, VOCs) from, e.g., shipping and fossil fuel resource extraction (Roiger et al., 2015). Such datasets would be particularly useful for the evaluation of atmospheric chemistry models and satellite products.

The changes in the abundance of anthropogenic aerosols and their precursors in Northern Eurasia have been extensive during the last decades (Granier et al., 2011), and this has almost certainly contributed to the very different regional warming patterns over these areas (e.g., Shindell and Faluvegi, 2009). The main anthropogenic aerosols in this context are primary carbonaceous particles, consisting of organic and black carbon, as well as secondary sulfate

particles produced during the atmospheric transport of sulfur dioxide. These species, as well
as nitrate, have also been found to dominate the aerosol composition at the ZOTTO site in
central Siberia (Mikhailov et al., 2015a, 2015b; Ryshkevich et al., 2015). These aerosols cause
large perturbations to the regional radiation budget downwind of major source areas in the
Northern Eurasian region, and the resulting changes in cloud properties and atmospheric
circulation patterns may be important even far away from these sources (Koch and Del Genio,
2010; Persad et al., 2012). In the snow-covered parts of Eurasia, long-range transported
aerosols containing black carbon and deposited onto snow tend to enhance the spring and
early-summer melting of the snow, with concomitant warming over this region (Flanner et al.,
2009; Goldenson et al., 2012; Meinander et al., 2015; Atlaskina et al., 2015).

The most important natural aerosol type over large parts of Eurasia is secondary organic

aerosol originating from atmospheric oxidation of biogenic volatile organic compounds
(BVOC) emitted by boreal forests and possibly other ecosystems. Studies conducted in the
Scandinavian part of the boreal zone indicate that new particle formation associated with
BVOC emissions is the dominant source of aerosol particles and cloud condensation nuclei
during summer time (Mäkelä et al., 1997; Kulmala et al., 2001; Tunved et al., 2006; Asmi et
al., 2011; Hirsikko et al., 2011). The production of secondary organic aerosols associated with
BVOC emissions has been estimated to induce large direct and indirect radiative effects over
the boreal forest zone (Spracklen et al., 2008; Tunved et al., 2008; Lihavainen et al., 2009,
2015; Scott et al., 2014). The few continuous measurement data sets from Siberia suggest
similarities in the frequency and seasonal pattern of new particle formation events between
Siberia and Nordic stations (Dal Maso et al., 2007; Arshinov et al., 2012; Asmi et al., 2015).
Measurements conducted at the ZOTTO site in central Siberia have shown that biogenic
secondary organic aerosols reach high concentrations in summer and dominate the aerosol
composition during this season (Mikhailov et al., 2015a,b; Ryshkevich et al., 2015). At this
site, however, new particle formation events are seen much less frequently than at the Nordic
stations (Heintzenberg et al., 2011). At present, relatively little is known about the overall
contribution of biogenic emissions to aerosol number or mass concentrations, or to the cloud
condensation nuclei budget, in Northern Eurasia.

Other important natural aerosol types in Northern Eurasia are sea spray, mineral dust,

and primary biogenic aerosol particles. Sea spray aerosol makes an important contribution to
the atmospheric aerosol over the Arctic Ocean and its coastal areas (Zábori et al., 2012, 2013),
and influences cloud properties over these regions (Tjernström et al., 2013). The climatic
effects of sea spray are expected to change in the future as a result of changes in the sea ice
cover and ocean temperatures (Struthers et al., 2011). Mineral dust particles affect regional
climate and air quality over large regions in Asia, especially during periods of high winds and
moderate precipitation. Mineral dust and primary biological aerosol particles (PBAP) particles
are also effective ice nuclei (Hoose and Möhler, 2012), and have the potential to influence the
radiative and other properties of mixed-phase cold clouds in the arctic-boreal regions. Over
Northern Eurasia, PBAP typically contribute more than 20% of PM2.5 organic aerosol mass
concentrations (Heald and Spracklen, 2009) and 25% of supermicron aerosol number
concentrations (Spracklen and Heald, 2014). Ice nucleation, in general, is one of the key
microphysical processes in the atmosphere that remain ill understood. However, a novel
theoretical approach (Laaksonen, 2015; Laaksonen and Malila, 2016) has been shown to be
superior to older theories in the case of water nucleation on solid surfaces, and it may open a
completely new avenue in the studies of atmospheric ice formation.

Satellites provide information about spatial distributions of the column-integrated

concentrations of aerosols (e.g., de Leeuw and Kokhanovsky, 2011; Andreae, 2009) and
various trace gases including ozone and its precursors (Burrows et al., 2011). These
atmospheric constituents are generally retrieved using passive instruments, which have good
sensitivity near the surface. However, retrieving information on the near-surface
concentrations of pollutants requires assumptions on their vertical distributions. For instance,
the retrieval of tropospheric ozone from satellite observations requires corrections for the high
concentrations in the upper troposphere and lower stratosphere. For aerosols, which can only
be retrieved in clear sky conditions, the situation may be complicated when disconnected
layers are present with different types of aerosols. A solution may be the retrieval of aerosol
vertical variation or the height of the aerosol layer using, e.g., active instruments (lidars), or
retrieval using spectrally-resolved observations in the Oxygen-A band (e.g., Hollstein and
Fisher, 2013), or instruments providing multiple viewing algorithms such as MISR (Nelson et
al., 2013) or AATSR (Virtanen et al., 2014). Another complication for aerosols may be the
vertical variation of the physical and chemical properties, which renders it difficult to obtain
closure between column and ground-based in situ measurements (Zieger et al., 2015 and
references cited therein). Nevertheless, good progress has been made in aerosol retrieval, and
column-integrated aerosol measurements (AOD) from satellites and ground-based
observations compare favorably (e.g., de Leeuw et al., 2015; Kolmonen et al., 2015).
Measurements of trace gases from space using wavelengths in the thermal infrared suffer from
low sensitivity in the lower troposphere (Pommier et al., 2010). All these factors may render
the comparison against local ground-based in-situ observations difficult, although a possible
way out could be the use of chemical transport models constrained by the satellite column
measurements (e.g., de Laat et al., 2009; Stavrakou, 2012; 2014), possibly together with sub-
orbital airborne measurements of relevant species. Satellite-measured AOD has been
successfully applied to obtain information on ground based aerosol mass concentrations
(PM2.5) (Xu et al., 2015; van Donkelaar et al, 2015). In addition, the use of multiple satellite
instruments, with different characteristics, is proposed to obtain more accurate information on
the transport of aerosols and trace gases and their vertical distribution (e.g., Naeger et al.,
2015). Recently, a technique has been demonstrated that makes it possible to derive CCN
concentrations at cloud base using remote sensing of cloud properties (Rosenfeld et al., 2016).
2.2.1.2. Large-scale pollutant transport and sources
Of particular interest is the pollutant transport to Arctic areas, where they can influence
the radiation budget and climate in various ways (Stohl, 2006; Warneke et al., 2009;
Meinander et al., 2013; Eckhardt et al., 2015). Model simulations suggest that European
emissions dominate Arctic pollutant burdens near the surface, with sources from North
America and Asia more important in the mid and upper troposphere (Monks et al., 2015). The
impact and influence of China and its polluted megacities on Arctic and boreal areas is a topic
of key importance, given recent and rapid Chinese industrialization. Inter-continental pollution
transport has also become of increased concern due to its potential influence on regional air
quality. The pollutant export from North America and Asia has been characterized by intensive
field campaigns (Fehsenfeld et al., 2006; Singh et al., 2006), but long-term research
approaches are lacking.
Emissions from forest fires (van der Werf et al., 2006; Sofiev et al., 2013) and from
agricultural fires in southern Siberia, Kazakhstan, and Ukraine (Korontzi et al., 2006) in spring
and summer are large sources of trace gases such as carbon monoxide (Nédélec et al., 2005;
Konovalov et al., 2014), as well as aerosol particles (Konovalov et al., 2015). Aerosols emitted
by forest fires are of particular interest, since the strength of this source type depends on both
climate change and human behavior (Pechony and Shindell, 2010), and since particles emitted
by these fires have potentially large radiative effects over Eurasia (Randerson et al., 2006). We
need comprehensive top-down emissions estimates, using inverse modeling constrained by
satellite observations, in order to provide quantitative information on the source strength of
aerosols and trace gases emitted by open fires.
Air pollution in monsoon Asia has two main characteristics. First, the total pollutant
emission rate from fossil fuel combustion sources is very high, leading to a high concentration
of primary and secondary pollutants in Asia, especially in eastern China and northern India.
Observations show that Asia is the only region where the concentrations of key pollutants,
such as nitrogen oxides (Richter et al., 2005; Mijling et al., 2013) and their end-product ozone
(Ding et al., 2008; Wang et al., 2009; Verstraeten et al., 2015), are still increasing. Second, in
addition to the anthropogenic fossil fuel combustion pollutants, monsoon Asia is also
influenced by intensive pollution from seasonal biomass burning and dust storms. For
example, intensive forest burning activities often take place in south Asia during spring and in
Siberia during summer, whereas intensive anthropogenic burning of agricultural straw takes
place in the north and east China plains. Dust storms frequently occur in the Taklimakan and
Gobi deserts in northwest China, and this dust is often transported over eastern China, southern
China, the Pacific Ocean and even the entire globe (Nie et al., 2014). After mixing with other
anthropogenic pollutants, biomass burning and mineral dust aerosols have been found to cause
complex interactions in the climate system (Ding et al., 2013; Nie et al., 2014).
2.2.2. Urban air quality

The northern Eurasian urban environments are characterized by cities with strong

anthropogenic emissions from local industry, traffic, and housing in Russia and China, and by
megacity regions with alarming air quality levels like those of Moscow and Beijing. Bad air
quality has serious health effects and damages ecosystems. In Beijing, for example,
concentrations of atmospheric fine particles have been found to be more than 10 times higher
than the safe level recommended by the World Health Organization (WHO) (Zheng et al.,
2015). Furthermore, atmospheric pollutants and oxidants play a central role in climate change
dynamics via their direct and indirect effects on global albedo and radiative transfer. A deeper
understanding of the unpredicted chemical reactions between pollutants and identification of
the most relevant feedbacks between air quality and climate at northern high latitudes and in
China is the most urgent task helping us to find practical solutions for more healthy air
(Kulmala, 2015).
In Siberian cities, the air quality is strongly linked to climatic conditions typical for
Siberia. Stable atmospheric stratification and temperature inversions are predominant weather
patterns for more than half of the year. This contributes to the accumulation of different
pollutants in the lowest layers of the atmosphere, thus increasing their impact on ecosystems
and humans. In addition to the severe climatic conditions, human impacts on the environment
in industrial areas and large cities continue to increase. In winter time, shallow and stably-
stratified planetary boundary layers (PBL) typical for northern Scandinavia and Siberia are
especially sensitive to even weak impacts and, therefore, deserve particular attention,
especially in the conditions of environmental and climate change (Zilitinkevich and Esau,
2009; Esau et al., 2012; Davy and Esau, 2014; Wolf et al., 2014 ; Wolf and Esau, 2014).
Unstably stratified PBLs interact with the free atmosphere mainly through turbulent ventilation
at the PBL upper boundary (Zilitinkevich, 2012). This mechanism, still insufficiently
understood and poorly modeled, controls the development of convective clouds, as well as
dispersion and deposition of aerosols and gases, which are essential features of heat waves and
other extreme weather events.
The worst air pollution episodes are usually associated with stagnant weather conditions
with a shallow PBL, which promotes the accumulation of intensively emitted pollutants near
the surface. The lower PBL is also influenced by the heavy pollution itself through its direct
or indirect effects on solar radiation and hence the surface sensible heat flux (e.g., Ding et al.,
2013b). The boundary layer - air pollution feedback will decrease the height of the PBL and
result in an even more polluted PBL (Ding et al., 2013b; Wang et al., 2014, Petäjä et al., 2016).
Therefore, considering the complex land surface types (city clusters surrounded by agricultural
areas) and pollution sources, improving our understanding of the associated feedbacks is very
important for forecasting extreme air pollution episodes and for long-term policymaking. In
order to understand this topic, more vertical measurements using aircraft, balloons, and remote
sensing techniques, as well as advanced numerical models including all relevant processes and
their couplings, are needed.
Planetary boundary layers are subject to diurnal variations, absorb surface emissions,
control microclimate, air pollution, extreme colds, and heat waves, and are sensitive to human
impacts. Very stable stratification in the atmosphere above the PBL prevents the compounds
produced by the surface fluxes or surface emissions from efficiently penetrating from the PBL
into the free atmosphere. This means that the PBL height and turbulent fluxes through the PBL
upper boundary control local features of climate and extreme weather events, such as the heat
waves associated with convection, or the strongly stable stratification events triggering the air
pollution (Zilitinkevich et al., 2015). This concept (equally relevant to the hydrosphere)
illustrates the importance of modeling and monitoring the atmospheric PBL height, which
varies from dozens to thousands of meters (Zilitinkevich, 1991; Zilitinkevich et al., 2007;
Zilitinkevich and Esau, 2009). To carry out a comprehensive inventory of the PBL height over
Northern Eurasia is urgently needed.
2.2.3. Atmospheric circulation and weather
The ongoing environmental change and its amplification in Northern Eurasia pose
special challenges to the prediction of weather-related hazards, and also to long-term impacts.
A key question is how the atmospheric dynamics (synoptic scale weather, boundary layer
characteristics) will change in Arctic and boreal regions. The recent changes in the Arctic sea-
ice have been much more rapid than models and scientists anticipated about ten years ago. The
role of the Arctic Ocean in the climate system and sea-ice changes have affected mid-latitude
weather and climate, with central and eastern Eurasia among the regions with strongest effects
(Vihma, 2014; Overland et al., 2015) (see section 2.3.1).
2.2.3.1. Atmospheric dynamics

The reliability of weather forecasts, and the extension of the time-range of useful

forecasts is needed for minimizing economic and human losses from extreme weather and
extreme weather related natural hazards. In Europe, this range is currently on average about
8−9 days (Bauer et al., 2015), which allows reliable early warnings to be issued for weather
related hazards, such as windstorms and extreme precipitation events with flash floods. The
time-range of useful forecasts has typically increased by a day per decade over the past three
decades (Uppala et al., 2005). In the Northern Eurasian region, improved predictions can be
used, for instance, for better prediction of thermal comfort conditions in Northern cities
(Konstantinov et.al, 2014). A strong urban heat island effect has already been observed in
urban areas of the Arctic with complex spatial and temporal structures (Konstantinov et al.,

2015).

Understanding of planetary boundary layer (PBL) processes is particularly important for
improving the weather predictions. The representation of boundary layer clouds, and their
further coupling to convection in stable conditions is not currently well understood.
Quantification of the behavior of the PBL over the Northern Eurasian region is needed in
analyses of spatial and temporal distribution of the surface fluxes, in predictions of
microclimate and extreme weather events, and in modeling clouds and air quality.

The development of diagnostic and modeling methods for aero-electric structures is

important for a study of both convective and electric processes in the lower troposphere
(Shatalina et al., 2005; 2007). Convection in the PBL leads to the formation of aero-electric
structures, manifested in ground-based measurements as short-period electric-field pulsations
with periods from several to several hundreds of seconds (Anisimov et al., 1999; 2002). The
sizes of such structures are determined by the characteristic variation scales of aerodynamic
and electrodynamics parameters of the atmosphere, including the PBL and surface-layer
height, as well as by the inhomogeneities in the ground (water) surface. Formed as a result of
convective processes and the capture of positive and negative charged particles (both ions and
aerosols) by convective elements (cells), aero-electric structures move with the airflow along
the Earth's surface. The further evolution of convective cells results, in particular, in cloud
formation.
2.2.3.2. Global Electric Circuit
The global electric circuit (GEC) is an important factor connecting the solar activity and
upper atmospheric processes with the Earth's environment, including the biosphere and
climate (Dolezalek et al. 1974, Singh et al. 2004). Thunderstorm activity maintains this circuit,
whose appearance is dependent on atmospheric conductance variations over a wide altitude
range. The anthropogenic impact on the GEC through aviation, forest fires and
electromagnetic pollution has been noted with great concern, and the importance of lightning
activity in climate processes has been recognized. The GEC forms because of two reasons: the
continuous operation of ionization sources, which provides an exponential growth of the
conductivity in the lower atmosphere, and the continuous operation of thunderstorm
generators, providing a high rate of electrical energy generation and dissipation in the
troposphere. Therefore, the GEC is influenced by both geophysical and meteorological factors,
and can serve as a convenient framework for the analysis of possible inter-connections
between atmospheric electrical phenomena and climate processes. Further exploration of the
GEC as part of the climate system studies, specifically its effect on the balance between the
Earth's ionosphere and global circuit, requires accurate modeling of the GEC stationary state
and its dynamics (Mareev 2010). Special attention should be paid to the observations and
modeling of generators (thunderstorms, electrified shower clouds, mesoscale convective
systems) in the global circuit.
2.3. Aquatic system - state of the art and future research needs
2.3.1. The Arctic Ocean in the climate system

The essential processes related to the interaction between the Arctic ocean and other

components of the Earth system include the air-sea exchange of momentum, heat, and matter
(e.g., moisture, aerosol, trace gases, $CO_2$, and $CH_4$), and the dynamics and thermodynamics of
sea ice. The most dramatic change in the Arctic Ocean has been the rapid decline of the sea
ice cover. Since the early the 1980s, the Arctic sea ice extent has decreased by roughly 50 %
in summer and autumn (Cavalieri and Parkinson 2012), while the winter sea ice thickness in
the central Arctic has decreased by approximately 50 % (Kwok and Rothrock 2009). Arctic
sea ice changes have serious teleconnections. Despite the warming climate, wintertime cold
spells in East Asia have become more frequent, stronger and longer lasting in this century
compared with the 1990s (Kim et al., 2014).  It also seems that the strong decline of the Arctic
sea ice has favored atmospheric pressure patterns that generate cold-air outbreaks from the
Arctic to East Asia (Mori et al., 2014; Kug et al., 2015; Overland et al., 2015). The reasons for
and the future evolution of the sea ice decline, as well as its effects on the ocean, atmosphere
and surrounding continents are among the current study topics of the Arctic climate system.
Other major issues include the role of the ocean in the Arctic amplification of climate change,
greenhouse gas exchange between the ocean, sea ice, and atmosphere, and aerosol budgets in
the marine Arctic (Smedsrud et al., 2013). The key question here is related to the changes of
sea ice extent and thickness, and to the terrestrial snow cover change.

Many of the processes considered to be responsible for the Arctic amplification of

climate warming are related to the ocean and sea ice (Döscher et al., 2014). Among these, the
snow/ice albedo feedback has received the most attention (e.g., Flanner et al., 2011). This
feedback is strongest when sea ice is replaced by open water, but it starts to play a significant
role already in spring when the snowmelt on top of sea ice begins. This is because of the large
albedo difference between dry snow (albedo about 0.85) and wet, melting, bare ice (albedo
about 0.40). More work is needed to understand quantitatively the reduction of snow/ice
albedo during the melting season, including the effects of melt ponds and pollutants in the
snow. Other amplification mechanisms related to the ocean include increased heat transports
from lower latitudes to the Arctic (Polyakov et al., 2010; Döscher et al., 2014) and fall-winter
energy loss from the ocean (Screen and Simmonds, 2010). Furthermore, the melting of sea ice
strongly affects evaporation, and hence the water vapor and cloud radiative feedbacks (Sedlar
et al., 2011), and the PBL thickness, which controls the sensitivity of the air temperature to
heat input into the PBL (Esau et al., 2012, Davy and Esau 2016). The relative importance of
the mechanisms affecting the Arctic amplification of climate warming are not yet well known
(See also Pithan and Mauritsen, 2014; Cohen et al., 2014).
The rapid decline of the Arctic sea ice cover has tremendous effects on navigation and
exploration of natural resources. To be able to predict the future evolution of the sea ice cover,
the first priority is to understand better the reasons, including the role of black carbon (see
Bond et al., 2013),  behind the past and ongoing sea ice evolution. Several processes have
contributed to the decline of Arctic sea ice cover, but the role of these processes needs better
quantification (Smedsrud et al., 2013; Vihma et al., 2014). Further studies are needed on the
impacts of changes in cloud cover and radiative forcing (Kay et al., 2008), atmospheric heat
transport (Kapsch et al., 2013) and oceanic heat transport (Döscher et al., 2014). In addition,
as the ice thickness has decreased, the sea ice cover becomes increasingly sensitive to the ice-
albedo feedback (Perovich et al., 2008). Other issues calling for more attention include the
reasons for the earlier onset of the spring melt (Maksimovich and Vihma, 2012), changes in
the phase of precipitation (Screen and Simmonds, 2011), and large-scale interaction between
the sea ice extent, sea surface temperature distribution, and atmospheric dynamics
(cyclogenesis, cyclolysis, and cyclone tracks) as discussed, e.g., by Outten et al., (2013).
In addition to thermodynamic processes, another factor affecting the sea ice cover in the
Arctic is the drift of sea ice. The momentum flux from the atmosphere to the ice is the main
driver of sea-ice drift, which is poorly represented in climate models (Rampal et al., 2011).
This currently hinders a realistic representation of sea-ice drift patterns in large-scale climate
models. Furthermore, the progressively thinning ice pack is becoming increasingly sensitive
to wind forcing (Vihma et al., 2012). In the future, research has to address the main processes
that determine the momentum transfer from the atmosphere to the sea ice, including the effects
of atmospheric stratification and sea ice roughness.
To understand better the links between the Arctic Ocean and terrestrial Eurasia, there is
a particular need to study the effects of Arctic sea ice decline on Eurasian weather and climate
(Section 2.2.3) Another poorly studied problem related to the Arctic Ocean is the role of sea
ice as a source of aerosol precursors, and in the gas exchange between the ocean and
atmosphere (Parmentier et al., 2013). Preliminary results of field studies at the drifting stations
North Pole 35 and 36 (Makshtas et al., 2011) showed that the shrinking sea ice cover could be
the reason for increasing $CO_2$ uptake from the atmosphere over the annual cycle, and for the
growth of the seasonal amplitude of $CO_2$ concentrations in the Arctic.
Climate models project that air temperatures and precipitation will increase over the
Arctic Ocean, and that this may have important effects on the structure of sea ice. Increased
snow load on a thinner ice may in the future cause flooding of seawater on ice in the Arctic,
which results in the formation of snow ice. Increased snow melt and rain, on the other hand,
results in increased percolation of water to the snow-ice interface, where it re-freezes, forming
super-imposed ice (Cheng et al., 2008). Snow ice and super-imposed ice have granular
structures, and differ thermodynamically and mechanically from the sea ice that currently
prevails in the Arctic.
The changes in the Arctic Ocean have opened some, albeit limited, possibilities for
seasonal prediction. These are mostly related to the large heat capacity of the ocean: if there is
little sea ice in the late summer and early autumn, this tends to cause large heat and moisture
fluxes to the atmosphere, favoring warm, cloudy weather in late autumn and early winter (Liu
et al., 2012; Stroeve et al., 2012). On the other hand, the reduction of the sea ice thickness may
decrease the possibilities for seasonal forecasting of ice conditions in the most favorable
navigation season in late summer - early autumn. This is because a thin ice is very sensitive to
unpredictable anomalies in the atmospheric forcing. For example, in August 2012 a single
storm caused a reduction of the sea ice extent by approximately 1 million km$^2$. The reduced
sea ice extent in the winter months has significant impacts on convective clouds. Observations
revealed a gradually increasing frequency of the convective cloud fields over Norwegian and
Barents Seas (Chernokulsky and Mokhov, 2012; Esau and Chernokulsky, 2015). The
unusually strong atmospheric convection and weaker virtual potential temperature inversions
create favorable conditions for the extreme Arctic cold outbreaks and meso-scale cyclones
known as Polar Lows (Kolstad et al., 2009).
It is vital to enhance routine observations, data assimilation techniques and prediction
models in order to properly monitor the physical state of the environment. Longer-term
impacts of the reduced ice cover are largely unknown, because the scientific community has
had only little time to create new knowledge on essential climate variables across the domain
(see section 2.3.1). To improve preparedness, new observational evidence is therefore needed
to reduce uncertainties in the system dynamics both on short and longer time-scales.
2.3.2. Arctic marine ecosystem
The ice cover of the Arctic Ocean is undergoing fast changes, including a decline of
summer ice extent and ice thickness (see 2.3.1). This results in a significant increase of the
ice-free sea surface in the vegetation season, and an increase in the duration of the growing
season itself. The key topic of future research is the joint effect of Arctic warming, ocean
freshening, pollution load, and acidification on the Arctic marine ecosystem, primary
production, and carbon cycle.
New ice-free areas of the Arctic Ocean could result in a pronounced growth of the annual
gross primary production (GPP), increased  phytoplankton biomass and a loss of ice-rich algae
communities associated with the low ice sheet surface (Bluhm et al., 2011). Progressive
increase of oil and natural gas drilling and transportation over the shelf areas will be escalating
the environmental changes of the Arctic marine ecosystems. Furthermore, there is a risk of
irreversible changes in marine Arctic productivity and key biogeochemical cycles, and the
potential for $CO_2$ absorption by marine ecosystem. Processes involving the Arctic may also
affect adjacent boreal areas.
We do not know how the climatically-induced increase in GPP and phytoplankton
biomass will influence the productivity of higher trophic levels of the Arctic ecosystem. In
typical Arctic ecosystems, the most important consumers are large-sized herbivorous
copepods, which have life cycles synchronized with the temperature as well as the seasonal
algae dynamics (Kosobokova, 2012). Another important consumer community are the small-
sized herbivorous copepods, which are important especially in shelf ecosystems. An increase
in the phytoplankton production in fall, together with an increase in the sea temperature, may
influence the populations of small-sized copepods, and increase their role in mass and energy
flow in the ecosystems. Our current understanding of the role of small copepods in the Arctic
ecosystems is limited (Arashkevich et al., 2010). An increase in surface water temperature
may "open the Arctic doors" for new species, and change the Arctic pelagic food webs, energy
flows, and biodiversity.
Increases in the Arctic sea temperature may lead to populations from neighboring
regions penetrating the Arctic ecosystem, changing the structure and functioning of native
ecosystems. For example, a 1.5 °C water temperature increase in the Bering Sea during the
mid-1970s allowed the Alaskan Pollock to penetrate the Arctic ecosystem, and occupy a place
as a keystone species for several years, supporting one of the world's largest regional fish
harvests (Shuntov et al., 2007). The Bering Sea ecosystem is very rich compared to the Arctic
ecosystems. Currently, we are not aware of food sources sufficient for supporting massive
invader populations even in case of climate-induced changes in ecosystems. However, the
appearance of aggressive new species even in low numbers may dramatically impact the
sensitive Arctic ecosystems and have effects on the future regulation of international fisheries
in the Arctic.
We have only recently begun to understand the processes that regulate freshwater-
marine ecosystem interactions in estuarine zones (Flint, 2010). The mechanisms determining
the impact of riverine waters over the Arctic shelves and the central deep-basin, and their
dependence on specific climatic forces, are still poorly understood. In order to determine the
impact of riverine waters, it is important to locate new flagship-stations or permanent
observation points in the estuaries of large Siberian rivers. The changing riverine discharge to
the Arctic shelves may amplify the impact of climate warming on the Arctic marine
ecosystems. Degradation of permafrost, soil erosion, changes in snow cover and summer
precipitation may all lead to changes in flood timing, and also to an increase in the amount of
fresh water and materials of terrestrial origin, including organic matter and nutrients, annually
delivered to the Arctic shelves, and further to the Arctic basin (Gustafsson et al., 2011).
Human-driven land use changes to drainage basins and associated river systems have the
potential to increase the speed of delivery of pollutants to the Arctic sea.
2.3.3. Lakes, wetlands and large-scale river systems
In the last decade, the combined effects of air pollution and climate warming on fresh-
water systems have received increasing attention (Skjelkvåle and Wright, 1998; Schindler et

al., 2001, Alcamo et al., 2002; Sanderson et al., 2006; Feuchtmayr et al., 2009; Sereda et al., 2011). It is important to understand the future role of Arctic boreal lakes, wetlands, and large river systems, including thermokarst lakes and running waters of all size, in biogeochemical cycles, and how these changes affect livelihoods, agriculture, forestry, and industry. The water chemistry of lakes without any direct pollution sources in the catchment area can be expected to reflect regional characteristics of water chemistry, as well as global anthropogenic processes, such as climate change and long-range air pollution (Müller et al., 1998; Moiseenko et al., 2001; Battarbee et al., 2005). The current ground-based stream flow-gauging network over the Norther Eurasian region does not provide adequate spatial coverage for many scientific and water management applications, including the verification of the land-surface runoff contribution to the recipients of intra-continental runoff. Special field laboratories, with joint observation and modeling capabilities in hydrometeorology, sedimentology, and geochemistry are needed to understand the spreading of tracers and pollutants as part of current and future global environmental fluxes.

The gradient in water chemistry from the tundra to the steppe zones in Siberia can provide insight into the potential effects of climate change on water chemistry. In the last century, long-range trans-boundary air pollution led to changes in the geochemical cycles of sulfur, nitrogen, metals, and other compounds in many parts of the world (Schlesinger, 1997; Vitousek et al., 1997a,b; Kvaeven et al., 2001; Skjelkvåle et al., 2001). Environmental pollution problems include also the waterborne spreading of nutrients and pesticides from local agricultural areas, heavy metals often originating from mining areas, and other elements and chemicals, such as persistent organic pollutants from urban and industrial areas. Shifts in downstream loads cause changes in river and delta dynamics. One example of important study area is the Selenga river basin, which is located in the center of Eurasia, extends from Northern Mongolia into southern Siberia (Russia), and has its outlet at Lake Baikal. The Selenga river

basin and Lake Baikal are located in the upstream part of the Yenisei River system, which
discharges into the Arctic Ocean. Lake Baikal has the largest lake volume in the world at about
23000 km$^3$ (comprising 20 % of all unfrozen freshwater in the world), hosts a unique
ecosystem (Granina, 1997), and is an important regional water resource (Garmaev and
Khristoforov, 2010; Brunello et al., 2006). There are numerous industries and agricultural
activities within the Selenga river basin, which affect the water quality of the lake and its
tributaries. Mining is well-developed in the region (e.g., Karpoff and Roscoe, 2005; Byambaa
and Todo, 2011), and heavy metals accumulate in biota and in sediments of the Selenga River
delta and Lake Baikal (Boyle et al., 1998; Rudneva et al., 2005; Khazheeva et al., 2006).
In addition to water chemistry, the role of aquatic systems as a net sink or source for
atmospheric $CO_2$ is presently under debate. When precipitation or other processes transport
large volumes of organic matter from land into nearby lakes and streams, the carbon of this
matter effectively disappears from the carbon budget of the terrestrial ecosystem (Huotari et
al., 2011). The enhanced decomposition of soil organic matter may significantly affect the
transport of terrestrial carbon to rivers, estuaries, and the coastal ocean. The contribution of
this process to the global and regional carbon budgets is unknown. Thus, the biological
processes taking place in the terrestrial ecosystem (e.g., photosynthesis, respiration, and
decomposition) and in the aquatic ecosystem are interlinked. The observed higher temperature
response of aquatic ecosystems compared to terrestrial ecosystems indicates that a substantial
part of the carbon respired or emitted from the aquatic system must be of terrestrial origin
(Yvon-Durocher et al., 2012). Long-term measurements carried out during all seasons in the
littoral zone of Lake Baikal showed that maximum $CO_2$ sink and emission rates are observed
in August and December (during the pre-ice period), respectively, and the total $CO_2$ flux from
the atmosphere into the littoral zone of Lake Baikal was estimated to be 3–5 g $CO_2$ m$^{-2}$
(Domysheva et al., 2013).
The Siberian lakes situated in tundra and forest-tundra zones are in general poorly
studied. In their natural state, their productivity is low, but their ecosystems are highly sensitive
to external influences. Profuse blooming of cyanobacteria is usually associated with urban and
industrial effluents and nutrient run-off. An assessment is needed of the impact of climate
change in the northern Eurasian region on eutrophication, accompanied by blooms of
cyanobacteria. Besides, the northern Eurasian region is characterized by thaw lakes, which
comprise 90% of the lakes in the Russian permafrost zone (Romanovskii et al., 2002). These
lakes, which are formed in melting permafrost, have long been known to emit $CH_4$. The latest
observations of the lakes in the permafrost zone of northern Siberia indicate that they are
releasing much more $CH_4$ into the atmosphere than previously thought. Rather than being
emitted in a constant flow, 95 % of $CH_4$ comes from random bubbling in disperse locations.
In coming decades, this could become a more significant factor in global climate change
(Walter et al., 2006).
One direct consequence of climate change is the explosive reproduction of toxic
cyanobacteria (*Nodularia, Microcystis, Anabaena, Aphanizomenon, Planktothrix*) and
diatoms (*Pseudo-nitzschia*) (Moore et al., 2008; Paerl and Huisman, 2009). These blooms
occur in ponds, lakes, reservoirs, and bays of the sea. Cyanobacteria and diatoms excrete
especially dangerous carcinogens and neurotoxins into the water. The toxicity of some
cyanotoxins exceeds the toxicity of currently banned warfare agents. Antidotes to these toxins
do not exist at the moment.
Water conservation has received an increasing attention in China, and multiple new
projects have been initiated recently. Especially the construction of water transfer, reservoir,
and irrigation schemes have received much attention, because the central and western regions
of China are suffering from water shortages. These projects are expected to improve water
usage and security, especially for agricultural activities, and to provide sufficient water
resources for local societies (Mu, 2014). In China, the river systems are dominated by rivers
flowing from the Tibetan plateau to the Pacific Ocean. The Yangtze is the longest river in
China, and flows from the Tibetan plateau to Shanghai. The Yellow river is the second longest
in China, and it is characterized by seasonal flooding which causes great economic and societal
losses. The Amur River forms the northern border with Russia. The Haihe River flows through
Beijing to Tianjin, and is under heavy stress from the highly populated and industrialized
capital metropolitan region. Only one river from China flows to the Arctic Ocean: the Ertix
River, which flows to the north through Kazakhstan, across Siberian Russia, finally joining
the Ob River, which flows to the Arctic Ocean.

2.4.   Social   system – state of the art and future research needs
2.4.1. Land use and natural resources

The fundamental large-scale task is to estimate how human actions such as land use

changes, energy production, the use of natural resources, changes in energy efficiency, and the
use of renewable energy sources will influence the environments and societies of the Northern
Eurasian region. For example, the industrial development of Siberia should be considered as
one of most important drivers of future land use and land cover changes in Russia. Siberia is
a treasure chest of natural resources of Russia, containing 85 % of its prospected gas reserves,
75 % of its coal reserves, and 65 % of its oil reserves. Siberia has more than 75 % of Russia's
lignite, 95 % of its lead, approximately 90 % of its molybdenum, platinum, and platinoids, 80
% of its diamonds, 75 % of its gold, and 70 % of its nickel and copper (Korynty, 2009).

During the 20[th] century, a considerable transformation of landscapes in the tundra and

taiga zones in northern Eurasia has occurred as a result of various industrial, socio-economic
and demographic processes, leading to the industrial development of previously untouched
territories (Bergen et al., 2013). This has led to a decrease in the rural population and, mostly
after the 1990s, to decrease in agricultural activities. There has also been a significant

reduction in agricultural land use, and its partial replacement by zonal forest ecosystems (Lyuri et al., 2010). According to recent estimates, the total area of abandoned agricultural land in Russia in the 1990s to 2010s is at about 57 million ha, of which 18 million ha have been restored by forests and 6 million ha of this are located in Asian Russia (Schepaschenko et al., 2015). As a result, these areas have become active accumulators of atmospheric $CO_2$ (Kalinina et al., 2009). These new forests ("substituting resources") could form the basis for sustainable development in these regions, in case relevant management programs for the forests re-established on abandoned lands are going to be implemented.

The dynamics of land cover, particularly forests, have been documented since 1961 when the results of the first complete inventory of Russian forests were published. According to official statistics, the area of forests in Asian Russia increased by around 80 million ha during 1961-2009, mostly before the middle of the 1990s. This large increase is explained by improved quality of forest inventories in remote territories, natural reforestation, mostly during the Soviet era as a result of forest fire suppression, and encroaching forest vegetation in previously non-forested land. Based on official statistics, the area of cultivated agricultural land in the region decreased by around 10 million ha between 1990 and 2009. After the year 2000, the forested area in Siberia decreased, mostly due to fire and the impacts of industrial transformations in high latitudes (Shvidenko and Schepaschenko, 2014). A critical decrease in the forest area has also been observed in the most populated areas with intensive forest harvesting particularly in the southern part of Siberia and the Far East. For example, in the Krasnoyarsk Krai, the total area of forests decreased by 5 %, while that of mature coniferous forests decreased by 25 %. Overall, the typical processes in these regions are a dramatic decline in the quality of forests, unsustainable use of forest resources, and insufficient governance and forest management in the region, including frequent occurrence of illegal logging, natural, and human-induced disturbances (Shvidenko et al., 2013).

Future land use and land cover changes will crucially depend on how successfully the

strategy of sustainable development of northern territories is developed and implemented. An

effective system for the adaptation of boreal forests to global change needs to be developed

and implemented in the region. An "ecologization" of the current practices of industrial

development of previously untouched territories would allow a substantial decrease in the

physical destruction of landscapes, and halt the decline of surrounding ecosystems due to air

pollution and water and soil contamination (Kotilainen et al., 2008).

The expected changes in the climate and environment will have multiple and

complicated impacts on ecosystems, with consequent land cover changes. The alteration of

fire regimes and the thawing of permafrost will intensify the process of "green desertification"

in large areas. Climate warming will have multiple effects on soil-vegetation-snow

interactions. For example, in a warmer climate, mosses and other vegetation grow faster,

providing a better thermal insulation of the permafrost in summer, and better feeding

conditions for reindeer. However, snow can also more easily accumulate on thicker vegetation,

thus protecting the deeper soil from cooling during the winter (Tishkov, 2012).

Both Russia's north and east possess abundant mineral resources (Korytnyi, 2009). The

resource orientation of northern and eastern Russia's economy, which has not changed for

centuries, increased in the post-Soviet period, and has been influenced primarily by the product

market. It is also expected that the natural resource development sector will continue to
dominate the economy in the majority of these territories for the next decades.
A crucial factor in greenhouse gas emission dynamics is the fuel balance. In Russia,
features of the fuel balance have led to an increased pollution. On average, specific emissions
in the northern and eastern cities of Russia, where coal accounts for most of the power
generation, are three times higher than in cities where power is generated mainly from gas or
fuel oil (Bondur, 2011a). The geographical location, undeveloped infrastructure, harsh
climate, and coal burning are the main reasons for increased levels of anthropogenic pollution
in these areas (Bondur and Vorobev, 2015; Bondur 2014). In small towns, low-capacity boiler
rooms are the main source of emissions. Usually, the lack of financial resources leads to the
use of low-quality coal and obsolete boilers. In the steppe zone of Asian Russia, Mongolia,
Kazakhstan, and Buryatia, the main source of emissions is the burning of harvest residues.

The dynamics of GHG emissions in Russia are largely determined by the economic

conditions of production. The economic crisis in 1990-1998 slowed down environmental
degradation to some extent: emissions generally decreased by 40 %. However, the underlying
environmental problems not only remained unresolved, but significantly deepened, and turned
into systemic problems. The most polluting industries were more resistant to the decline in
production. Technological degradation took place, cleaning systems were eliminated, and
production shifted to part-time, leading to inefficient capacity utilization. Significant amounts
of pollution continued to be emitted from the domestic sector. Emissions decreased in most
regions of the country, and in 83% of the cities, but much more slowly than production. As a
result, the specific emissions (per product cost at comparable prices) had grown by the end of
the 1990s in all categories of cities, except cities with more than 1 million inhabitants
(Bityukova et al., 2010). All this can cause negative impacts on ecosystems. For example,
there are about 2 million ha of technogenic deserts around Norilsk. Norilsk is probably the
biggest smelter in the world, and produces more than 2 million tons of pollutants per year
(Groisman et al., 2013).
2.4.2. Natural hazards
2.4.2.1. Extreme weather and fire occurrence
The frequency and intensity of weather extremes have increased substantially during the

last decades in Europe, Russia, and China. Further acceleration is expected in the future (IPCC
2013. The evolving impacts, risks, and costs of weather extremes on population, environment,
transport, and industry have so far not been properly assessed in the Northern latitudes of
Eurasia. New knowledge is needed for improving the forecasting of extreme weather events,
for understanding the effect of wildfires on radiative forcing and atmospheric composition in
the region, for estimating the impacts of weather extremes on major biogeochemical cycles,
and for understanding the effects of disturbances in forests on the emissions of BVOC and
VONs (volatile organic nitrogen) (Bondur, 2011b, 2015; Bondur, Ginsburg, 2016). How do
changes in the physical, chemical and biological state of the different ecosystems and the
inland, water, and coastal areas affect the economies and societies in the region, and vice
versa?

The number of large hydrometeorological events in Russia that cause substantial

economic and social losses has increased by more a factor of two from 2001 to 2013 (State
Report, 2014). The main hazards are related to atmospheric processes on various temporal and
spatial scales, including strong winds, floods and landslides caused by heavy precipitation, and
fires caused by drought and extreme temperatures. High temperatures and long droughts can
substantially decrease the productivity and cause high dieback in dark coniferous forests.
Hurricanes occur fairly often in the forest zone. For example, a hurricane destroyed about
78000 ha of forest in the Irkutsk region in July 2004 (Vaschuk and Shvidenko, 2006).
However, there are no reliable statistics on many types of natural hazards.

In order to build scenarios of the future frequency and properties of weather-related

hazards, one should first analyze the atmospheric mechanisms behind the circulation structures
responsible for these hazards: the cyclones related to strong winds and heavy precipitation and
the anticyclones related to drought and fires episodes. Studying the cyclone/anticyclone tracks,
frequency and intensity can provide a statistical basis for understanding the geographical
distribution and properties of the major atmospheric hazards and extremes (e.g., Shmakin and
Popova, 2006). For future climate projections, atmospheric hazards and extremes should be
interpreted from the viewpoint of cyclone/anticyclone statistics, and possible changes in the
cyclone/anticyclone geography and frequency should be analyzed.
Fires are the most important natural disturbances in the boreal forests. Fires strongly
determine the structure, composition and functioning of the forest. Each year, about $0.5-1.5$
% of the boreal forest burns. Since boreal forests cover 15 % of the Earth's land surface, this
is a significant area (Kasischke, 2000; Conard et al., 2002; Bondur, 2011b, 2015). Climate
change already substantially impacts fire regimes in northern Eurasia. More frequent and
severe catastrophic (mega-) fires have become a typical feature of the fire regimes. Such fires
envelope areas of up to a hundred thousand hectares within large geographical regions, lead to
the degradation of forest ecosystems, decrease the biodiversity, may spread to usually
unburned wetlands, cause large economic losses, deteriorate life conditions and health of local
populations, and lead to "green desertification", which is an irreversible transformation of the
forest cover for long periods (Shvidenko and Schepaschenko, 2013, Bondur, 2011b, 2015).
Megafires also lead to specific weather conditions over the affected areas that are comparable
in size to large-scale pressure systems (~30 million ha and more). The annually burned area in
the Russian territory was estimated to be $8.2\pm0.8\cdot10^6$ ha during 1998-2010, and about two
thirds of this area consisted of boreal forests. For this period, the fire carbon balance (total
amount of carbon in the burnt fuel) was estimated to be $121\pm28$ Tg C year$^{-1}$ (Shvidenko et al.,
2011). Current model projections suggest that the number of fires will double by the end of
this century. The extent of catastrophic fires escaping from the control and fire intensity are
projected to increase. Due to increased severity of fire and deeper soil, carbon emissions from
fires are predicted to increase by a factor of 2 to 4 (Gromtsev, 2002; Malevsky-Malevich et
al., 2008; Flanningan et al., 2009; Shvidenko et al., 2011). During and after fires, significant
changes take place in the forest ecosystems, including the soil. These changes include: (i) a
significant amount of biomass is combusted, and large amounts of carbon and nitrogen are
released to the atmosphere in the form of carbon dioxide and other gases or particles (Harden
et al., 2000; Andreae and Merlet, 2001; Kaiser et al., 2012; Konovalov et al., 2014; Kulmala
L. et al., 2014); (ii) fire alters the microbial community structure in the soil, as well as the
structure of the vegetation (Dooley and Treseder, 2012; Sun et al., 2015); (iii) fires determine
the structure of the vegetation, succession dynamics and the fragmentation of forest cover, tree
species composition, and the productivity of boreal forests (Gewehr et al., 2014) and (iv) fire
is one of the crucial drivers controlling the dynamics of the carbon stock of boreal forests
(Jonsson and Wardle, 2010; Köster et al., 2014).
Disturbances resulting from fire, pest outbreaks and diseases also have substantial effects
on the emissions of BVOCs and volatile organic nitrogen compounds (Isidorov, 2001), and
consequently on atmospheric aerosol formation. The acceleration of fire regimes will also
affect the amount of black carbon in the atmosphere, and thus has an effect on the albedo of
the cryosphere.
2.4.2.2. Permafrost degradation and infrastructures
The degradation of permafrost will cause serious damage both to infrastructure and to
ecosystems and water systems in the Northern Eurasian region. This includes, for example,
damage to pipelines and buildings, deformation of roads and railroads in Russia, Mongolia
and China, variations in the ion distribution in soil water in young and ancient landslides,
cryogenic landslides, spatial and temporal changes of grass and willow vegetation, saline water
accumulation in local depressions of the permafrost table, and formation of highly saline lenses
of ground water called 'salt traps'.
Due to the large extent of permafrost-covered areas in northern Eurasia (for ecosystem
effects, see sections 2.1.1 and 2.1.2), there are numerous infrastructural issues related to
possible changes in the thickness and temperature of the frozen part of the subsurface, and thus
in the mechanical soil properties. Climate change -induced changes in the cryosphere are
probably among the most dramatic issues affecting the infrastructure in northern Eurasia, as
this infrastructure is literally standing on permafrost. Moreover, an interesting coupling may
be related to the decreasing ice-cover of the Arctic Ocean, which results in increased humidity
and precipitation on the continent, and thus a further thickening and longer duration of the
annual snow cover. Snow is a good thermal insulator, and influences the average ground
surface temperature, thus playing a potentially important role in speeding up the thawing of
permafrost.
The increased risk of damage to local infrastructure, such as buildings and roads, can
cause significant social problems, and exerts pressure on the local economies. Thawing
permafrost is structurally weak, and places a variety of infrastructure at risk. For example, the
failure of buildings, roads, pipelines, or railways can have dramatic environmental
consequences, as seen in the 1994 breakdown of the pipeline to the Vozei oilfield in northern
Russia, which resulted in a spill of 160,000 tons of oil - the world's largest terrestrial oil spill
(United Nations Environment Program, 2013). Maintenance and repair costs related to
permafrost thaw and degradation of infrastructure in northern Eurasia have recently increased,
and will most probably increase further in the future. This is an especially prominent problem
in discontinuous permafrost regions, where even small changes in the permafrost temperature
can cause significant damage to infrastructure. Most settlements in permafrost zones are
located on the coast, where strong erosion places structures and roads at risk. After damage to
the infrastructure, local residents and indigenous communities are often forced to relocate.
This can cause changes in, or even disappearances of, local societies, cultures, and traditions
(United Nations Environment Program, 2013).
2.4.2.3. Changing sea environments and the risk of accidents in coastal regions
In northern Eurasia, from the eastern part of the Barents Sea to the Bering Sea, the
permafrost is located directly on the seacoast. In many of these coastal permafrost areas, sea
level rise and continuing permafrost degradation leads to  significant coastal erosion, and to
the possibility of a collapse of coastal constructions such as lighthouses, ports, houses, etc. In
this region, the sea level rise is coupled to the permafrost degradation in a complex way, and
should be focused on in future studies.
Understanding and measuring artificial radionuclides in marine ecosystems is needed
for improving emergency preparedness capabilities, and for developing risk assessments of
potential nuclear accidents. The awareness of the general public and associated stakeholders
across the region should also be raised concerning the challenges and risks associated with
nuclear technologies, environmental radioactivity, and emergency preparedness. The current
state of radioactive contamination in terrestrial and marine ecosystems in the European Arctic
region will be studied by examining environmental samples collected from Finnish Lapland,
Finnmark, and Troms in Norway, the Kola Peninsula, and the Barents Sea. The results will
provide updated information on the present levels, occurrence and fate of radioactive
substances in the Arctic environments and food chains. The results will also allow us to
estimate where the radioactive substances originate from, and what risks they may pose in case
of accidents.
Annual expeditions for sample collection are needed for the development of models to
predict the distribution of radionuclides in the northern marine environment, and for the
assessment of the current state of radioactive contamination in marine ecosystems in the
European Arctic region. In view of recent developments and increased interests in the
European Arctic region for oil and gas extraction, special attention needs to be given to the
analysis of norms (naturally occurring radioactive materials) in order to understand current
levels. The future focus should be put on atmospheric modeling, and on the assessment of
radionuclide distributions in the case of accidents leading to the release of radioactive
substances to the environment in the European Arctic region. This includes the assessment of
nuclear accident scenarios for dispersion modeling.

### 2.4.3. Social transformations

Climate and weather strongly affect the living conditions, mostly in the Eastern part of the Northern Eurasian societies, influencing people's health, incidence of diseases and adaptive capacity. The vulnerability of societies, including their adaptive capacity, varies greatly depending on both their physical environment, and on their demographic structure and economic activities. There is a need to analyzes the scientific background and robustness of the adaptation and mitigation strategies (AMS) of the region's societies, and their resilience capacity, with special emphasis on the forest sector and agriculture. The future research needs are in understanding in what ways populated areas are vulnerable to climate change; how their vulnerability can be reduced and their adaptive capacities improved; what responses should be identified to mitigate and adapt to climate changes.

Health issues are also important in multidisciplinary studies of northern Eurasia, as the living conditions of both humans and livestock are changing dramatically. Short-lived climate forcers (SLCF), such as black carbon, ozone, and aerosol particles, are important players in both air quality and Arctic climate change and their impacts are not yet quantified. Black carbon has a special role when designing future emission control strategies, since it is the only major aerosol component whose reduction is likely to be beneficial to both climate and human health. These changes can be expressed through complex parameters combining the direct effects of, e.g., temperature and wind speed, with indirect effects of several climatic and non-climatic factors such as the atmospheric pressure variability, or the frequency of unfavorable weather events, such as heat waves or strong winds. During the last decades, living conditions in northern Eurasia have generally improved, but with a significant regional and seasonal variation (Zolotokrylin et al., 2012).

Both northern and eastern Eurasia have small and diminishing populations, mainly due to the migration outflow started in the 1990s due to severe and unfavorable living conditions combined with changing state policies with respect to the development of the northern

territories. This reversed the previous long-standing pattern of migration inflow. The combination of outflow and natural population decrease (with some regional exceptions in several ethnic republics and autonomous regions (*okrugs*) with oil and gas industry) led to a steady population decline in most regions in northern and eastern Russia from 1990s. In the post-soviet period, the population of eastern Russia decreased by 2.7 million, while the population of Russia's Arctic zone decreased by nearly by one third (500 000 people), in contrast to the majority of the world's Arctic territories (Glezer, 2007a, 2007b). The population change in northeastern Russia was particularly remarkable: the Chukotka autonomous okrug lost 68 % of its population, the Magadan Oblast lost 59 %, and the Kamchatka Krai lost 33 %.

Geographical and ethnic factors influence the demography and settlement pattern in the region. Geographical factors include environmental conditions and the mixture of urban and rural territories. Areas with a large proportion of indigenous people employed in traditional nature management were exposed to relative small post-soviet transformations in the 1990's and 2000's. In contrast, the largest transformations occurred in areas with a larger proportion of Russian people and developed mining industries. The differences in the transformations between settlements with predominantly indigenous and predominantly Russian populations are evident. For example, in the Chukotka Autonomous Okrug, the former remained mostly intact, with only small decreases in population, while the latter disappeared entirely or were significantly depopulated (Litvinenko, 2012; 2013).

When assessing the impacts of climate change and other environmental changes on human societies, it should be taken into account that the urban environments in northern Eurasian cities and towns situated in the less favored regions are currently incapable of mitigating unfavorable impacts. The impact of climate parameters, such as temperature (including seasonal, weekly, and daily cycles, and extreme values), strong winds, snowfall, snowstorms, and precipitation should be investigated. Both the frequency and the duration of

weather events should be considered. These climate parameters influence human health,
incidence of diseases, adaptation potential, and economic development in general.
Furthermore, it is important to explore the interactions between the environmental change and
post-soviet transformations of natural resource utilization in northern Eurasia in order to assess
the complexity of their socio-ecological consequences at regional and local levels (Litvinenko,
2012; Tynkkynen, 2010). The population dynamics of the northern Russian regions in 1990-
2012, and the linkage between intra-regional differences in population dynamics, spatial
transformations of natural resources utilization, and ethnic composition of the populations
should be clarified. It would be desirable to develop an "early warning system" for the timely
mitigation of the negative socio-ecological effects of both environmental changes, and changes
in the availability of natural resources as well as accident like leakages in gas and oil pipelines.
Such systems would be useful for federal, regional, and local authorities, as well as for local
communities.
It should also be taken into account that the majority of the world's ethnic groups are
small and engaged in culturally specialized methods of subsistence, so any change in their
immediate environment may lead to their traditional way of life becoming unsustainable.
These changes may be due to rising sea levels, warming seawater, melting ice cover, thawing
permafrost, flooding rivers, changing rain patterns, or moving vegetational zones. These are
direct effects of climate change and environmental deterioration on ethnodiversity. However,
even more threatening are the indirect effects. The immediate environment of small ethnic
groups is often vulnerable to the adverse impact of majority populations representing
governments and nations. The effects of climate change may lead to a rapid and massive
transfer of majority populations to areas previously inhabited by small ethnic groups.

3. From process studies towards system understanding and quantification of feedbacks of Arctic-boreal regions

The system understanding helps us to understand the behavior of feedbacks between the land, atmosphere, aquatic, and societal/economic systems. To be able to provide a system understanding, we need to understand the individual processes, and based on process understanding we are then able to quantify different biogeochemical cycles. Via biogeochemical cycles, the energy and matter flows are linked to a wider system context, which enables us to analyze the feedback phenomena. Feedbacks are essential components of our climate system, as they either increase or decrease the changes in climate-related parameters in the presence of external forcings (IPCC, 2013).

The effects of climate change on biogeochemical cycles are still inadequately understood, and many feedback mechanisms are difficult to quantify (Arneth et al., 2010; Kulmala et al., 2014). They are related to, for example, the coupling of carbon and nitrogen cycles, permafrost processes and ozone phytotoxicity (Arneth et al., 2010), or to the emissions and atmospheric chemistry of biogenic volatile organic compounds (Grote and Niinemets, 2008; Mauldin et al., 2012), subsequent aerosol formation processes (Kulmala et al., 2004; Tunved et al., 2006; Kulmala et al., 2011a; Hirsikko et al., 2011) and aerosol-cloud interactions (McComiskey and Feingold, 2012; Penner et al., 2012; Rosenfeld et al., 2014).

The northern Eurasian Arctic-boreal geographical region covers a wide range of interactions and feedback processes between humans and natural systems. Humans are acting both as the source of climate and environmental changes, and as recipient of their impacts. The PEEX research agenda is addressing the most relevant research topics related to the process dynamics in the land, atmospheric, aquatic, and society systems relevant to Northern regions. PEEX also aims to quantify the range of emissions and fluxes from different types of ecosystems and environments and links to ecosystem productivity (see also Su et al., 2011; Kulmala and Petäjä, 2011; Bäck et al., 2010). This new knowledge helps us to obtain a holistic

view on the changes in biogeochemical cycles and feedbacks in the future Arctic-boreal system
(Fig 4). PEEX will also to take into consideration that there may exist previously unknown
sources and processes (Su et al., 2011; Kulmala and Petäjä, 2011; Bäck et al., 2010).

Holistic representations of feedback loops potentially relevant to Arctic-boreal systems

have been given by Charlson et al., (1987), Quinn and Bates (2011), and by Kulmala et al.,
(2004; 2014). The "CLAW" hypothesis ("CLAW" acronym refers to Charlson, Lovelock,
Andreae and Warren) connects the ocean biochemistry and climate via a negative feedback
loop involving cloud condensation nuclei production due to dimethylsulfoniopropionate
(DMSP) and DMS biosynthesis by marine phytoplankton (e.g., Quinn and Bates, 2011;
Ducklow et al., 2001; O'Dowd et al., 2004; de Leeuw et al., 2011; Malin et al., 1993; O'Dowd
and de Leeuw, 2007). The COBACC (COntinental Biosphere-Aerosol-Cloud-Climate)
hypothesis suggests two partly overlapping feedbacks that connect the atmospheric carbon
dioxide concentration, ambient temperature, gross primary production, biogenic secondary
organic aerosol formation, clouds, and radiative transfer (Kulmala et al., 2004; 2014; also see
section 2.1.1.). The quantification of these feedback loops under changing climate is crucial
for reliable Earth system modelling and predictions.

In the context of the COBACC feedback loop, the key large-scale research questions are

the changing cryospheric conditions and consequent changes in ecosystem feedbacks affecting
the Arctic-boreal climate system and weather. Furthermore, we should estimate the net effects
of various feedback effects (CLAW, COBACC) on land cover changes, photosynthetic
activity, GHG exchanges, BVOC emissions, aerosol and cloud formation, and radiative
forcing at regional and global scales. In our analysis, we should also take in account the
urbanization processes and social transformations (see section 2.4.3), which are changing the
regional climates. In this task, we should also study the key gaps of the biogeochemical cycles.

3.1. Hydrological cycle

Climate change may profoundly affect most of the components of the hydrological cycle, giving rise to positive or negative feedbacks (Fig 5). While variations in the hydrological cycle often take place at regional or local scales, they can also give rise to large-scale or even global changes. Knowledge of the hydrological cycle in general and particularly related to permafrost is crucial for predicting the resilience and transformation of forest ecosystems coupled with permafrost (Osawa et al., 2009).

In addition to permafrost processes, another important issue in high latitudes is precipitation. Precipitation is a critical component of the hydrological cycle, having a great spatial and temporal variability. The lack of understanding of some precipitation-related processes, combined with the lack of global measurements of sufficient detail and accuracy, limits the quantification of different components of the hydrological cycle like precipitation, evapotranspiration, CCN formation etc. This is especially true in high-latitude regions, where observations and measurements are particularly sparse, and processes poorly understood.

Recent retrievals of multiple satellite products for each component of the terrestrial water cycle provide an opportunity to estimate the water budget globally (Sahoo et al., 2011) (Fig. 5). Global precipitation is retrieved at very high spatial and temporal resolution by combining microwave and infrared satellite measurements (Sorooshian et al., 2000; Kummerow et al., 2001; Joyce et al., 2004; Huffman et al., 2007). Large-scale estimates of global precipitation have been derived by applying energy balance, process, and empirical models to satellite-derived surface radiation, meteorology, and vegetation characteristics (e.g., Mu et al., 2007; Su et al., 2007; Fisher et al., 2008; Sheffield et al., 2010). The water storage change component can be obtained from satellite data, and the water level in lakes and large-scale river systems can be estimated from satellite altimetry with special algorithms developed for terrestrial waters (Berry et al., 2005; Velicogna et al., 2012; Troitskaya et al., 2012; 2013).

3.2. Carbon cycle

It is not clear how future climate will modify incoming terrestrial net primary production (NPP) and outgoing (e.g., heterotrophic soil respiration (HSR)) carbon fluxes to and from terrestrial ecosystems. It is likely that the transformation of Russian forests is a tipping element for the climate system by end of the century over huge areas, even though uncertainties in such forecasts are significant (Gauthier et al., 2015). The role of boreal and Arctic lakes and catchment areas in carbon storage dynamics is poorly quantified (Fig. 6).

The terrestrial biosphere is a key regulator of atmospheric chemistry and climate via its carbon uptake capacity (Arneth et al., 2010; Heimann and Reichstein, 2008). The Eurasian area holds a large pool of organic carbon both within the above- and below-ground living biota, in the soil, and in frozen ground, stored during the Holocene and the last ice age. The area also contains vast stores of fossil carbon. According to estimates of carbon fluxes and stocks in Russia made as part of a full carbon account by the land-ecosystem approach (Shvidenko et al., 2010a; Schepaschenko et al., 2011; Dolman et al., 2012), terrestrial ecosystems in Russia served as a net carbon sink of 0.5-0.7 Pg(C) per year during the last decade. Forests provided above 90 % of this sink. The spatial distribution of the carbon budget shows considerable variation, and substantial areas, particularly in permafrost regions and in disturbed forests, display both sink and source behavior. The already clearly observable greening of the Arctic is going to have large consequences on the carbon sink in the upcoming decades (Myneni et al., 1997; Zhou et al., 2001), although future predictions are uncertain. The Net Ecosystem Carbon Budget (NECB) or Net Biome Production (NBP) are usually a sensitive balance between carbon uptake through forest growth, ecosystem heterotrophic respiration, and carbon release during and after disturbances such as fire, insect outbreaks, or weather events such as exceptionally warm autumns (Piao et al., 2008; Vesala et al., 2010). This balance is delicate, and for example in the Canadian boreal forest the estimated net carbon

balance is close to carbon neutral due to fires, insects, and harvesting cancelling the carbon
uptake from forest net primary production (Kurz and Apps, 1995; Kurz et al., 2008). ). Long-
term measurements of the concentrations of $CO_2$ and other carbon gases at selected sites,
especially using tall towers such as the ZOTTO tower, will be essential for constraining the
large-scale carbon fluxes in the PEEX region (Heimann et al., 2014).
Plant growth and carbon allocation in boreal forest ecosystems depend critically on the
supply of recycled nutrients within the forest ecosystem. In the nitrogen-limited boreal and
Arctic ecosystems, the biologically available nitrogen ($NH_4$ and $NO_3$) is in short supply,
although the flux of assimilated carbon below ground may stimulate the decomposition of
nitrogen-containing soil organic matter (SOM), and the nitrogen uptake of trees (Drake et al.,
2011; Phillips et al., 2011). The changes in easily decomposable carbon could enhance the
decomposition of old SOM (Kuzyakov, 2010; Karhu et al., 2014), and thus increase the
turnover rates of nitrogen in the rhizosphere, with possible growth-enhancing feedbacks on
vegetation (Phillips et al., 2011).
Arctic warming is promoting terrestrial permafrost thaw and shifting hydrologic flow
paths, leading to fluvial mobilization of ancient carbon stores (Karthe et al., 2014). Observed
permafrost thaw acts as a significant and preferentially degradable source of bioavailable
carbon in Arctic freshwaters, which is likely to increase as permafrost thaw intensifies, causing
positive climate feedbacks in response to on-going climate change (Mann et al., 2015).
Significant differences in fluvial carbon input between headwaters and downstream reaches of
large Arctic catchments (Yenisey and Lena) have been identified, but the problem is very
poorly explained yet. At the same time, the fluvial export by the largest rivers is considered to
be an order of magnitude less than coastal erosion in the East Siberian Arctic Shelf (Semiletov
et al, 2011). The Lena's particulate organic carbon export is estimated to be two orders of
magnitude less than the annual input of eroded terrestrial carbon onto the shelf of the Laptev
and East Siberian seas.

Although inland waters are especially important as lateral transporters of carbon, their

direct carbon exchange with the atmosphere, so-called outgassing, has been recognized to be
a significant component in the global carbon budget (Bastviken *et al.*, 2011; Regnier et al.,
2013). In the boreal pristine regions, forested catchment lakes can vent *ca.* 10 % of the
terrestrial NEE (Net Ecosystem Exchange), thus weakening the terrestrial carbon sink (Huotari
*et al.*, 2011). There is a negative relationship between the lake size and gas saturation, and
especially small lakes are relatively large sources of $CO_2$ and $CH_4$ (e.g., Kortelainen et al.,
2006; Vesala, 2012). However, on a landscape level, large lakes can still dominate the GHG
fluxes. Small lakes also store relatively larger amounts of carbon in their sediments than larger
lakes. The role of lakes as long-term sinks of carbon, and simultaneously as clear emitters of
carbon-containing gases, is strongly affected by the physics of the water column. In lakes with
very stable water columns and anoxic hypolimnion sediments, carbon storage is especially
efficient, but at the same time, these types of lakes emit $CH_4$. In general, the closure of
landscape-level carbon balances is virtually impossible without studying the lateral carbon
transfer processes (Pumpanen et al., 2014), and the role of lacustrine ecosystems as GHG
sources/sinks. Besides lakes, these studies should include rivers and streams, which could be
even more important than lakes as transport routes of terrestrial carbon and as emitters of
GHGs (Huotari et al, 2013). In addition, the role of VOC emissions as a part of the carbon
budget needs to be quantified.
3.3. Nitrogen cycle

Nitrogen is the most abundant element in the atmosphere. However, most of the

atmospheric nitrogen is in the form of inert $N_2$, which is unavailable most for plants and
microbes, and can only be assimilated into terrestrial ecosystems through biological $N_2$
fixation (Canfield et al., 2010). Only cryptogamic covers and certain organisms living in
symbiosis with plants are capable of nitrogen fixation, making nitrogen the main growth-
limiting nutrient in terrestrial ecosystems (Elbert et al., 2012; Lenhart et al., 2015). Human
perturbations to the natural nitrogen cycle have, however, significantly increased the
availability of nitrogen in the environment (Fig. 7). These perturbations mainly stem from the
use of fertilizers in order to increase crop production to meet the demands of the growing
population (European Nitrogen Assessment, 2010), although atmospheric nitrogen deposition
may also play a significant role in some areas. The increased use of fertilizer nitrogen, and
consequent perturbations in nitrogen cycling, also cause severe environmental problems such
as eutrophication of terrestrial and aquatic ecosystems, atmospheric pollution, and ground
water deterioration (European Nitrogen Assessment, 2010).
Emission of reactive nitrogen (NO, $NO_2$, HONO, ammonia, amines) from soils (Su *et*
*al.*, 2011; Korhonen *et al.*, 2013), fossil fuel burning, and other sources links the nitrogen cycle
to atmospheric chemistry and secondary aerosol formation in the atmosphere. There are
indications that emissions of $N_2O$ from the melting permafrost regions in the Arctic may
significantly influence the global $N_2O$ budget and hence contribute to the positive radiative
forcing by greenhouse gases (Repo et al., 2009; Elberling et al., 2011).
In natural terrestrial ecosystems, nitrogen availability limits ecosystem productivity,
linking the carbon and nitrogen cycles closely together (Gruber and Galloway, 2008). The
increasing temperatures due to climatic warming accelerate nitrogen mineralization in soils,
leading to increased nitrogen availability and transport of reactive nitrogen from terrestrial to
aquatic ecosystems. This perturbed and accelerated nitrogen cycling may lead to large net
increases in the carbon sequestration of ecosystems (Magnani et al., 2007). The large surface
area of boreal and Arctic ecosystems implies that even small changes in nitrogen cycling or
feedbacks to the carbon cycle may be important on the global scale (Erisman et al., 2011). For
instance, increased atmospheric nitrogen deposition has led to higher carbon sequestration in
boreal forests (Magnani et al., 2007). However, the feedback mechanisms from increased
perturbations of the nitrogen cycle may change the dynamics of the emissions of other
greenhouse gases, hence complicating the overall effects. For instance, the stimulated carbon
uptake of forests due to increased atmospheric nitrogen deposition can largely be offset by the
simultaneously increased soil $N_2O$ emissions (Zaehle et al., 2011). In the Arctic, the melting
permafrost may lead to high emissions of $N_2O$ (Repo et al., 2009; Elberling et al., 2010), which
may significantly influence the global $N_2O$ budget.
Understanding the processes within the nitrogen cycle, the interactions of reactive
nitrogen with the carbon and phosphorus cycles, atmospheric chemistry and aerosols, as well
as their links and feedback mechanisms is therefore essential in order to fully understand how
the biosphere affects the atmosphere and the global climate (Kulmala and Petäjä, 2011).
3.4. Phosphorus cycle
Phosphorus (P) is, together with nitrogen (N), one of the limiting nutrients for terrestrial
ecosystem productivity and growth, while in marine ecosystems, phosphorus is the main
limiting nutrient for productivity (Whitehead and Crossmann, 2012). The role of P in nutrient
limitation in natural terrestrial ecosystems has not been recognized as widely as that of N
(Vitousek et al., 2010).
In the global phosphorus biogeochemical cycle, the main reservoirs are in continental
soils, where phosphorus in mineral form is bound to soil parent material, and in ocean
sediments (Fig.8). Sedimentary phosphorus originates from riverine transported material
eroded from continental soils. The atmosphere plays a minor role in the phosphorus cycle, and
the phosphorus cycle does not have a significant atmospheric reservoir. Atmospheric
phosphorus mainly originates from aeolian dust, sea spray, and combustion (Wang et al.,
2014). Gaseous forms of phosphorus are scarce, and their importance for atmospheric
processes is unknown (Glindemann et al., 2005).
Southwestern Siberian soils have lately been reported to contain high concentrations of
plant-available phosphorus (Achat et al., 2013), which may enhance the carbon sequestration
of the ecosystems if they are not too limited by nitrogen.   In soils, phosphorus is found mainly
in mineral form and bound to the soil parent material such as apatite minerals. The amount of
phosphorus in the parent material is a defining factor for phosphorus limitation, and the
weathering rate determines the amount of phosphorus available for ecosystems. In ecosystems,
most of the available phosphorus is in organic forms (Achat et al., 2013; Vitousek et al., 2010).
In ecosystems growing on phosphorus-depleted soils, the productivity is more likely to be
nitrogen-limited in early successional stages, and gradually shift towards phosphorus
limitation as the age of the site increases (Vitousek et al., 2010). In freshwater ecosystems,
excess phosphorus leads to eutrophication, which has ecological consequences, such as the
loss of biodiversity due to changes in physicochemical properties and in species composition
(Conley et al., 2009). Due to the scarcity of studies focusing on ecosystem phosphorus cycling,
the effects of climate change on physicochemical soil properties and phosphorus availability,
and the interactions of the phosphorus cycle with the cycles of carbon and nitrogen, are largely
unknown.
3.5. Sulfur cycle
Sulfur is released naturally through volcanic activity, as well as through weathering of
the Earth's crust. The largest natural atmospheric sulfur source is the emission of dimethyl
sulfide (DMS) from oceanic phytoplankton (Andreae, 1990). DMS is converted to sulfur
dioxide ($SO_2$), sulfuric acid ($H_2SO4$) and methyl sulfonic acid (MSA) via gas-phase oxidation.
However, human activities have a major effect on the global sulfur cycle via vast emissions of
$SO_2$ from fossil fuel burning and smelting activities. The main sink of $SO_2$ is oxidation to
sulfuric acid in both gas and liquid phases, and subsequent removal from the atmosphere via
precipitation and dry deposition.
Global anthropogenic $SO_2$ emissions are predicted to decrease significantly by the year
2100 (IPCC, special report on emissions scenarios, SRES, 2000). Emissions in Europe and
North America started to decrease already in the 1970s, but this decrease is still overwhelmed
on a global scale by increasing emissions in eastern Asia and other strongly developing regions
of the world (Smith et al., 2011). The current global anthropogenic $SO_2$ emissions are about
120 Tg per year, with Europe, the former Soviet Union and China together responsible for
approximately 50 % (Smith et al., 2011). Global natural emissions of sulfur, including DMS,
are significantly smaller (a few tens of Tg per year; Smith et al., 2001). Anthropogenic
emissions dominate especially over the continents. The main sources of $SO_2$ are coal and
petroleum combustion, metal smelting, and shipping, with minor contributions from biomass
burning and other activities.
$SO_2$ emissions in Eurasia have a large spatial variability. Smelters in the Russian Arctic
areas emit vast amounts of $SO_2$, significantly affecting the regional environment. In 2007
Blacksmith Institute experts estimated, that the smelter complexes in Norilsk, with annual
emissions of 2 Tg are alone responsible for more than 1.5 % of global $SO_2$ emissions.
However, the emissions from the smelters in the Kola Peninsula, while still remarkably high,
have decreased significantly during the past decades (Paatero et al., 2008), thus altering the
impact of human activities on the regional climate and environment. In general, existing
anthropogenic activities are slowly becoming more sulfur-effective and less polluting.
However, the emergence of new sulfur-emitting activities and infrastructures partially
counteracts this development.
The behavior of future changes in $SO_2$ emissions in the PEEX research area is uncertain.
In northern Eurasia, natural resources like fossil fuels, metals, minerals, and wood are vast,
and their utilization is becoming more and more attractive due increasing demand. This will
most likely lead to an increase in human activities (e.g., mining, oil drilling, shipping) in this
area (e.g., Smith, 2010, and references therein). For example, sulfur emissions in China
increased rapidly until 2006, and then decreased by 9.2 % to 30.8 Tg in 2010 due to the wide
application of flue-gas desulfurization (FGD) equipment in power plants (Lu and Zhang 2011).
Sulfur emissions in Europe have decreased significantly during the last decades (Jones and
Harrison 2011).
Most of the natural and anthropogenic $SO_2$ is removed from the atmosphere by liquid-
phase oxidation to $H_2SO_4$, and subsequent precipitation. In areas with high sulfur loadings,
acid rain leads to acidification of soils and waters (Fig. 9). The main final sink of sulfur is the
oceans. A fraction of $SO_2$ is oxidized to $H_2SO_4$ in the gas phase in a reaction chain initiated by
the reaction of $SO_2$ with the hydroxyl radical, OH. Especially in forested areas of Eurasia,
reactions of $SO_2$ with a second important oxidant type, the stabilized Criegee intermediates
originating from biogenic VOC emissions, also produce significant amounts of $H_2SO_4$
(Mauldin et al., 2012). Gas-phase sulfuric acid plays a key role in the Earth's atmosphere by
triggering secondary aerosol formation, thus connecting anthropogenic $SO_2$ emissions to
global climate via aerosol-cloud interactions. Particles containing sulfuric acid, or sulfate, are
also connected with air quality problems and human health deterioration. Understanding the
spatial and temporal evolution of $SO_2$ emissions in northern Eurasia, along with atmospheric
sulfur chemistry, is crucial for understanding and quantifying the impacts of anthropogenic
activities and $SO_2$ emissions on air quality, acidification, as well as on regional and global
climate.

4. From system understanding to mitigation and adaptation strategies and decision making

Climate change and weather extremes are already affecting the living conditions of
Northern Eurasian societies. The vulnerability of the Northern environments and societies,
including their adaptive capacity and buffering thresholds, varies greatly depending on their
current and future physical environment as well as their demographic structure and economic
activities. The PEEX program as a whole is built on four pillars, namely (i) research, (ii)
research infrastructure, (iii) impact on society, and (iv) knowledge transfer and capacity
building. The scientific outcome of the first two pillars will be addressing the future state of
the physical environment and its interactions and feedbacks with the demographic structure
and economic activities in the Arctic boreal system. Periodic PEEX assessments will be
delivered for constructing mitigation and adaptation strategies of the Northern societies and
for use of regional and governmental decision-making. The PEEX approach is applicable to
China, when taking into account the specific geographical, climatological, and social
characteristics of that region.
The integrative approach of the PEEX first two pillars provides both analytical and
operational answers to our research questions, which can be utilized in solving interlinked
grand challenges using pillars iii) and iv). These will also contribute to the Earth System
sciences (ESS) questions as a whole (see ESS questions: Schellnhuber et al., 2004). The
implementation of the PEEX research agenda starts with process studies in the frame of three
main topics determined for the land, atmosphere, aquatic, and social systems of the Northern
Eurasian region. The research approach is designed to answer the analytical questions on the
major dynamical patters and feedback loops relevant to Earth system science in the Northern
context. The PEEX program has defined altogether 12 large-scale research questions for the
12 main topics in the Northern Eurasian domain (Kulmala et al., 2016). At the same time,
PEEX adheres to several operational ESS questions, including "what level of complexity and
resolution have to be achieved in Earth System modelling?", "what are the best techniques for
analyzing and predicting irregular events?", "what might be the most effective global strategy
for generating, processing, and integrating relevant Earth system datasets?", and "what are the
most appropriate methodologies for integrating natural science and social science
knowledge?" (Schellnhuber et al., 2004).
In terms of the level of complexity and resolution in Earth System modelling, PEEX
builds on a multi-scale modelling and observation approach originally introduced by Kulmala
et al., (2009). PEEX will construct its own multi-scale modelling platform (Lappalainen et al.,
2014). In terms of generating, processing and integrating relevant Earth system datasets, a
detailed conceptual design of the PEEX research infrastructure (RI) will include a concept
design of a coherent in-situ observation network, coordinated use of remote sensing
observations and standardized and harmonized data procedures, as well as a data system. One
of the first tasks of PEEX-RI is to fill in the observational gap in atmospheric in-situ and
ground base remote sensing data in the Northern Eurasia, especially in Siberia. This approach
is based on the coordination of existing observation activities (Alekseychik et al., 2016), but
also making plans for a new infrastructure needed. PEEX-RI development will be largely
based on the SMEAR (Station for Measuring Ecosystem-Atmosphere Relations) concept
(Kulmala et al., 2016), which has been developed by the University of Helsinki Division of
Atmospheric Sciences together with Division of Forest Ecology starting from 1995 (Hari and
Kulmala, 2005; Hari et al., 2016). The SMEAR concept provides a state-of-the-art foundation
for establishing a PEEX observation system to be integrated into the global GEOSS data
system. Furthermore, detailed design of greenhouse gas, aerosol, cloud, and trace gas
measurements, and observation of biological activity will find synergies with the major
European land-atmosphere observation infrastructures, such as ICOS (a research infrastructure
to determine the greenhouse gas balance of Europe and adjacent regions), ACTRIS (aerosols,
clouds, and trace gases research infrastructure), GAW (Global Atmospheric Watch), and
AnaEE (the experimentation in terrestrial ecosystem research).

PEEX is interested in developing methodologies for integrating natural science and

social science knowledge as part of the operational Earth sustainable system questions
(Schellnhuber et al., 2004). The first-priority task in this case is to establish an integrated
geographical information background (Ribeiro et al., 2009; Hunt and Sanchez-Rodriquez,
2009; Shvidenko et al., 2010; Skryzhevska et al., 2015). A common information background
would be the first step serving the development of a common language of integrated studies.
For example, we need spatially and temporally explicit descriptions of terrestrial ecosystems,
landscapes, atmosphere, and hydrosphere. A common information background would be a
unified base for the PEEX modelling platform and for the development of integrated modelling
clusters, which could combine ecological, economic, and social dimensions. It could provide
a historical background for the future trajectories of land cover, state and resilience of
ecosystems, stability of landscapes, and dynamics of environmental indicators of environment.
The already exiting Integrated Land Information System could be utilized here for combining
all historical knowledge about the region and all scientific results obtained by past, current,
and future studies across the region (e.g., Schepaschenko et al., 2011; Shvidenko and
Schepaschenko, 2014).

In addition to data services, PEEX is developing procedures for integrating and linking

natural science and social science knowledge and data. As one example, we need to analyze
data on emission sources together with population health risk factors, environment pollution,
food security, drinking water quality, changes in the spreading areas of infectious diseases,
and changes in the general epidemiological situation (Bityukova and Kasimov, 2012;
Malkhazova et al., 2013). Via novel multidisciplinary data interfaces and data procedures, we
are able to connect satellite observations with inverse modeling, provide fast updates to
emission inventories, estimate the emission for the climate models, and, in the end, provide
climate and air quality scenarios and the storylines of the future development of the arctic-
boreal region (Fig. 10 ).
In terms of strategic questions of the ESS, such as "what is the optimal mix of adaptation
and mitigation measures to respond global change?" or "what is the structure of an effective
and efficient system of global and development of institutions?", PEEX is an active player in
creating direct contacts with the stakeholders, so that its scientific information and services
will receive an optimal impact on decision making. Furthermore, the PEEX approach endorses
the                    Earth                    System                    Manifesto.
(https://www.atm.helsinki.fi/peex/images/manifesti_peex_ru_hub2.pdf),    which    addresses
three strategic tasks: (i) construction of novel observation systems, (ii) finding consensus
addressing necessary mitigation and adaptation actions in different parts of the world, and (iii)
operational  prerequisites  for  technological  development  to  moderate  the  global  change
towards the sustainable Earth System. In this framework, PEEX will work closely with
influential organizations, such as the Intergovernmental Panel for Climate Change (IPCC)
delivering PEEX assessment of Arctic-boreal region, the Future Earth acting as an Arctic-
Boreal Hub, and the Digital Earth via demonstrating novel methods for integrating in-situ data
with satellite observations.

Acknowledgements
The preparatory work of the PEEX approach, started in 2012, has been mainly based on
the in-kind contribution of several European, Russian, and Chinese research institutes. The
work presented here would not have been possible without the collaboration of the PEEX
meetings participants. In 2012-2015 we have organized five PEEX meetings in Helsinki (2012,
2015), Moscow (2013), Hyytiälä (2013), and Saint Petersburg (2014), and the first PEEX
Science conference in Helsinki, Finland in February 2015. PEEX has also been  active in a
frame of international coordination activities and has as such been listed as GEOSS - Gold
region project, IGBP-iLEAPS Arctic and boreal regional node, Digital Earth, Arctic Council
– SAON Task, one of the main collaborators of the International Eurasian Academy of
Sciences (IEAS) and Future Earth Arctic-boreal hub.

In addition, we would like acknowledge the support and funding from the following

bodies: Academy of Finland Centre of Excellence (grant no.  272041), "International Working
Groups, Markku Kulmala" Grant by Finnish Cultural Foundation; ICOS 271878, ICOS-
Finland No. 281255, ICOS-ERIC No. 281250, No. 259537, 218094, 255576, 286685, 280700
and 259537 funded by Academy of Finland; Beautiful Beijing project funded by TEKES, In
GOS DEFROST and CRAICC (no 26060) and Nordforsk CRAICC-PEEX (amendment to
contact 26060) funded by Nordforsk. "European-Russian Centre for cooperation in the Arctic
and Sub-Arctic environmental and climate research" (EuRuCAS, grant 295068); Erasmus+
CBHE project ECOIMPACT 561975 funded by EU 7$^{th}$ Framework Program.

We thank many Russian projects, which have been contributed PEEX and have been

granted by national funding organizations: Russian Mega-Grant No.  11.G34.31.0048
(University of Nizhny Novgorod),  Russian Ministry of Education and Science Grants (unique
project identifiers RFMEFI58614X0004 and RFMEFI58314X0003,  ISR "AEROCOSMOS",
2014-2016); Russian Science Foundation projects No. 15-17-20009 (University of Nizhny
Novgorod) and No. 15-17-30009 (Faculty of Geography, Moscow University, 2015-2018),
RFBR Grant № 14-05-91759 (ISR "AEROCOSMOS", 2014-2016), Russian Science
Foundation projects No. 14-47-00049 (A.M. Obukhov Institute of Atmospheric Physics RAS,
2014-2016), No. 11.37.220.2016 (St.Petersburg State University) and No. 14-27-00083 (the
Geochemical foundations of PEEX), No.15-5554020 (SINP Moscow University, 2015-2016)
and No. № 14-17-00096 (Saint-Petersburg State University, 2014-2016).    We also

acknowledge Presidium of the Russian Academy of Sciences (Program No. 4); the Branch of Geology, Geophysics and Mining Sciences of RAS (Program No. 5); interdisciplinary integration projects of the Siberian Branch of the Russian Academy of Science No. 35, No. 70 and No. 131; Russian Foundation for Basic Research (Grants No. 14-05-00526, No. 14-05-00590, No. 14-05-93108) and Russian Science Foundation project No. № 14-17-00096 (Saint-Petersburg State University, 2014-2016). AARI  thanks CNTP 1.5.3.2 and 1.5.3.4 of Roshydromet. Institute of Geography thanks Russian Academy of Sciences and Russian Science Foundation (project № 14-27-00133).

University of Tartu together with University of Life Sciences, Estonia, acknowledge the European Commission through European Regional Fund (the "Internationalization of Science Programme" project INSMEARIN (10.1-6/13/1028) and the "Estonian Research Infrastructures Roadmap" project Estonian Environmental Observatory, 3.2.0304.11-0395). The University of Tartu, Estonia, acknowledges the institutional research funding IUT20-11 of the Estonian Ministry of Education and Research for it's support for the development of SMEAR Estonia at Järvselja. Nansen Center, Norway, acknowledges the International Belmont Forum project "Anthropogenic Heat Islands in the Arctic - Windows to the Future of the Regional Climates, Ecosystems and Societies" No.: 247468/E10.

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

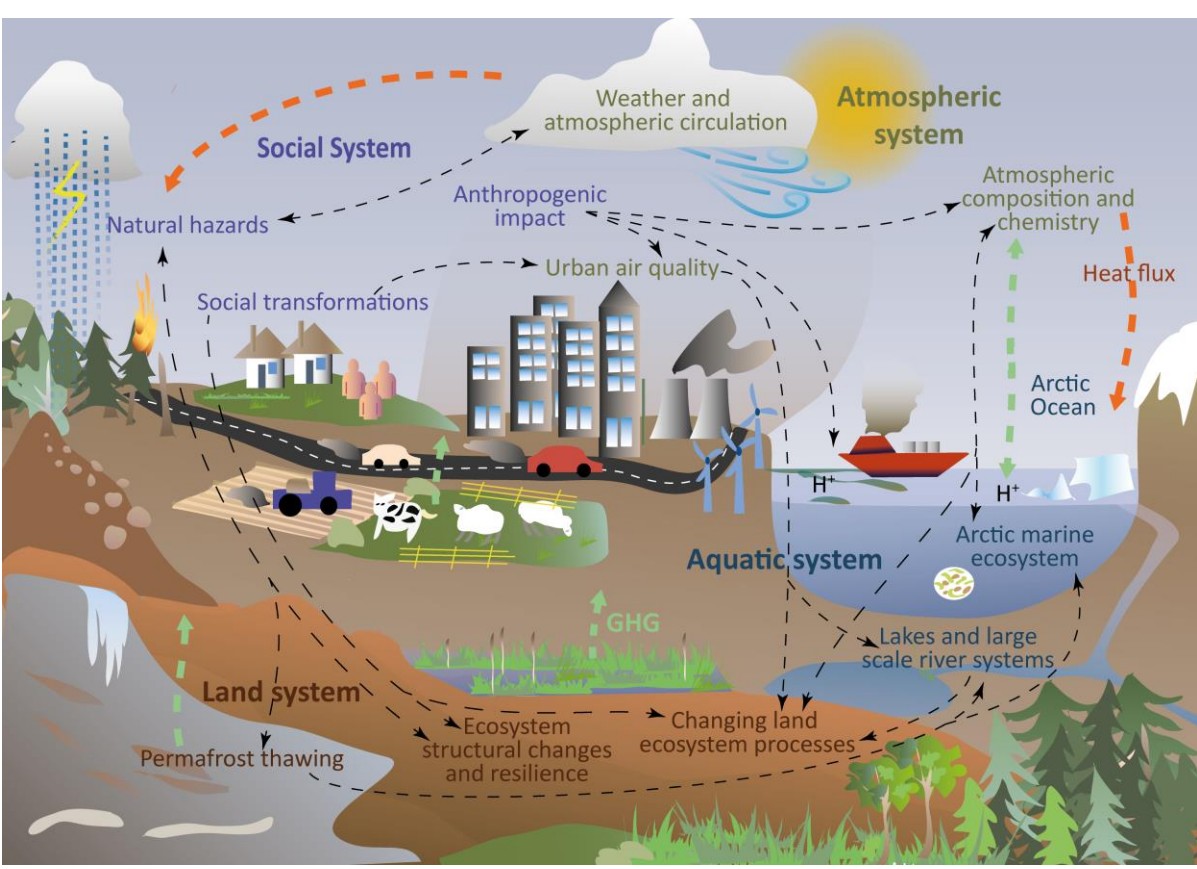


Figure 1: The thematic research areas relevant to the Northern Eurasian land system include
Land-Topic-1 "changing ecosystem processes", Land-Topic-2 "ecosystem structural changes
and resilience" and Land-Topic--3 "risk areas of permafrost thawing". For the atmospheric

system they are Atmosphere-topic-1 "atmospheric composition and chemistry", Atmosphere-topic-2 "Urban air quality", are Atmosphere-topic-3, "atmospheric circulation and weather", for the aquatic system they are Aquatic-topic-1 "Arctic Ocean in the climate system", Aquatic-topic-2 "maritime ecosystems", Aquatic-topic-3 "Lakes and large river systems" and for the social system they are Society-topic-1 "natural resources and anthropogenic activities", Society-topic-2 "natural hazards" and Society-Topic-3 "social transformations".

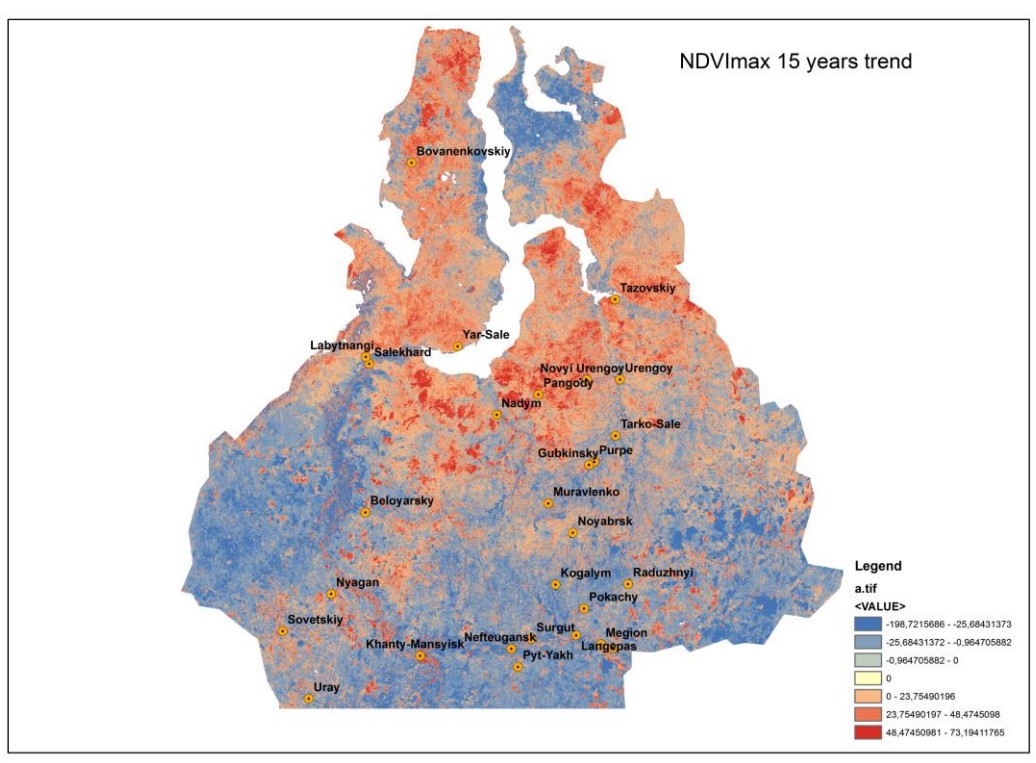

Figure 2: Linear trends in the annual maximum Normalized Difference Vegetation Index (NDVI) obtained from analysis of the MODIS 0.25 km data product for 2000-2014 over the North-Western Siberia region in Russia. The trends are given in the NDVI changes per 15 years. The yellow colors show the decreasing NDVI, which corresponds to decreasing biological production; the blue colors show the increasing NDVI. More detailed analysis of the trends is given in Esau et al. (2016).

2995

2996

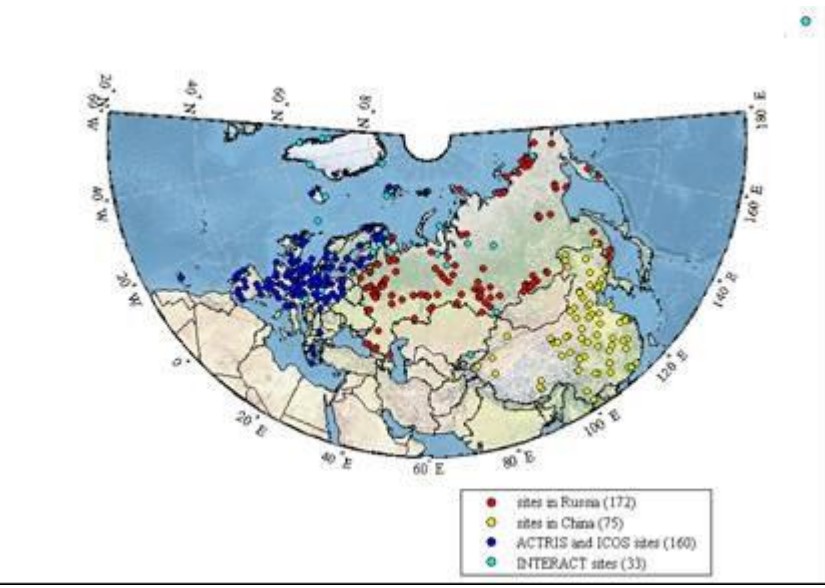

2997

Figure 3: Map showing the existing ACTRIS (Aerosols, clouds and trace gases Research Infrastructure Network) and ICOS (Integrated Carbon Observations System) stations in Europe (blue), stations making atmospheric and/or ecosystem measurements in Russia (red), INTERACT (International Network for Terrestrial Research and Monitoring in the Arctic) stations in Russia (light blue) and China Flux stations in China (yellow). However, all of these stations need certain upgrades.


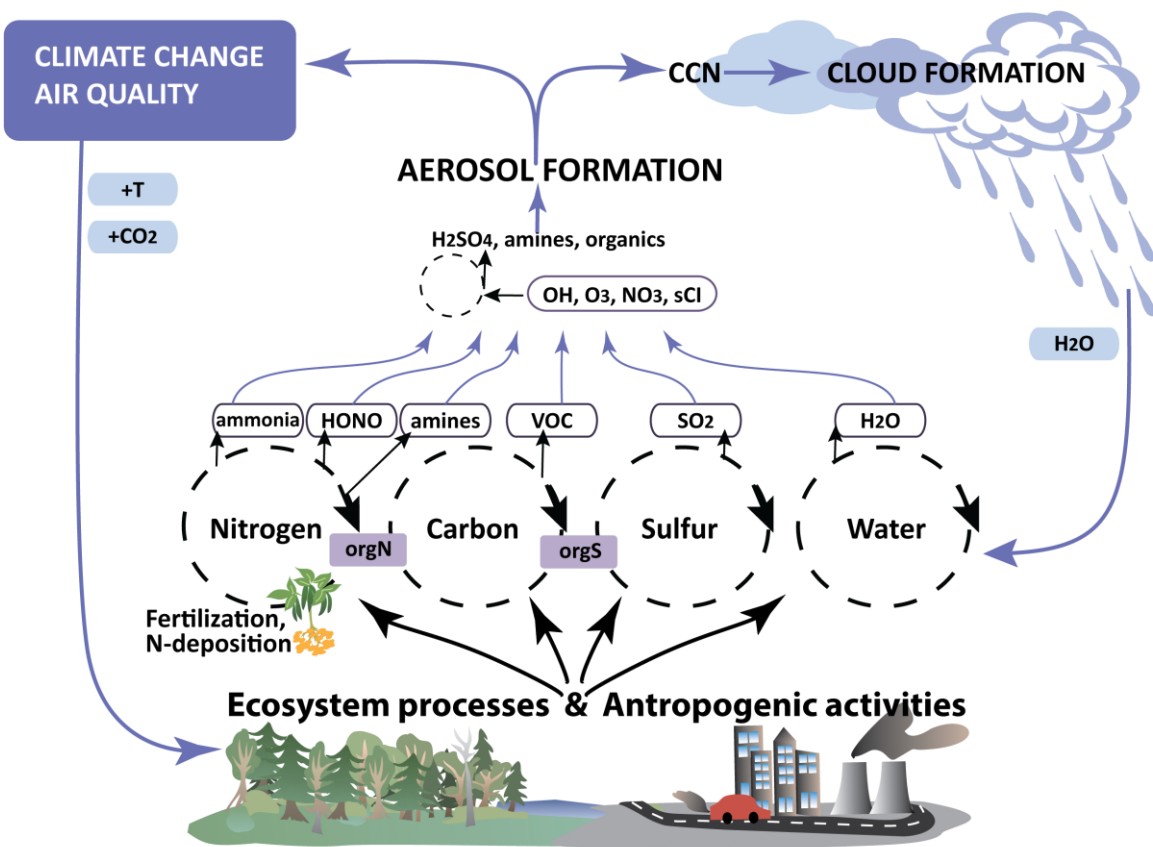



Figure 4: In urban and industrialized regions, the process understanding of biogeochemical
cycles includes anthropogenic sources, such as industry and fertilizers, as essential parts of the
biogeochemical cycles.

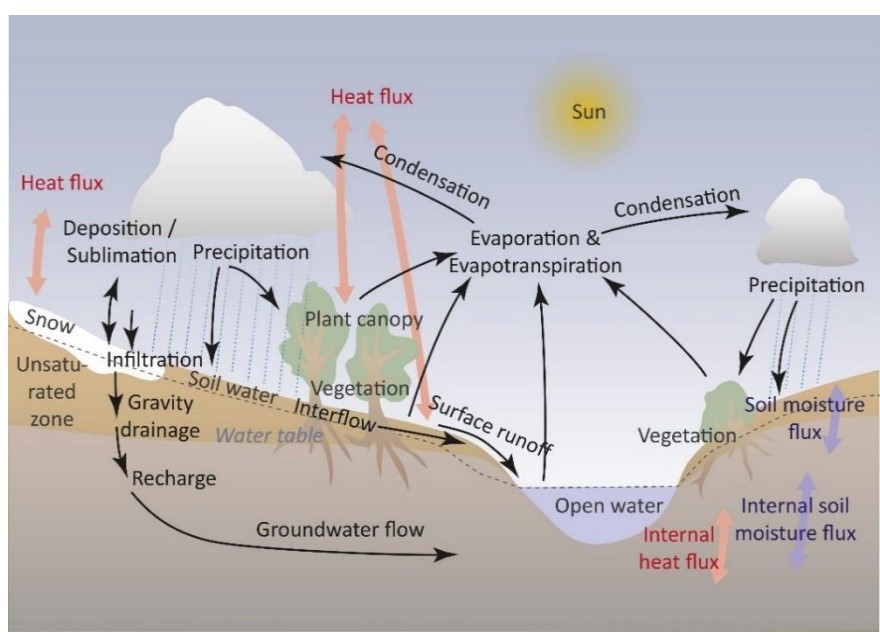


Figure 5: Hydrological cycle schematics.

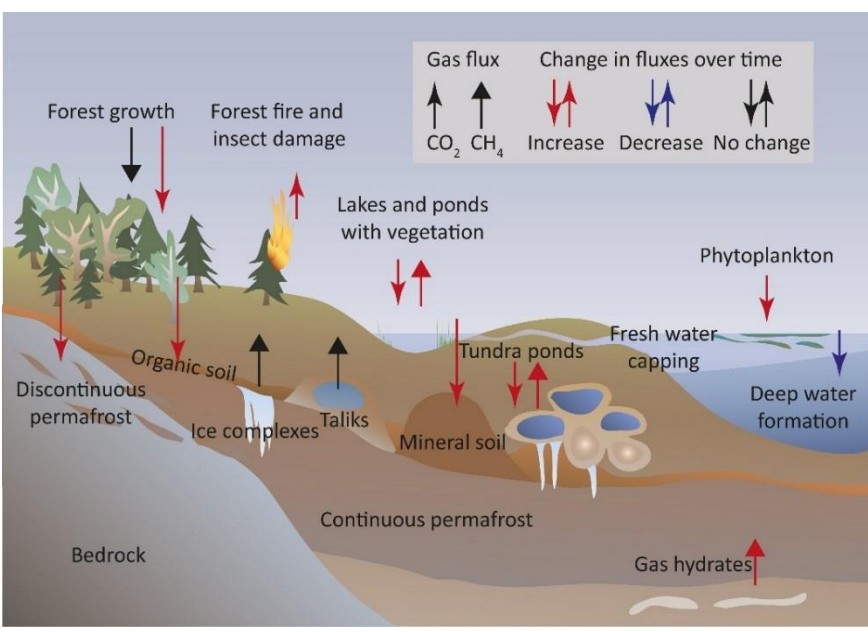


Figure 6: Carbon cycling in the Arctic will change as the climate warms. Figure after ACIA,
2004. (Impacts of a Warming Arctic: Arctic Climate Impact Assessment (ACIA) Overview
Report).




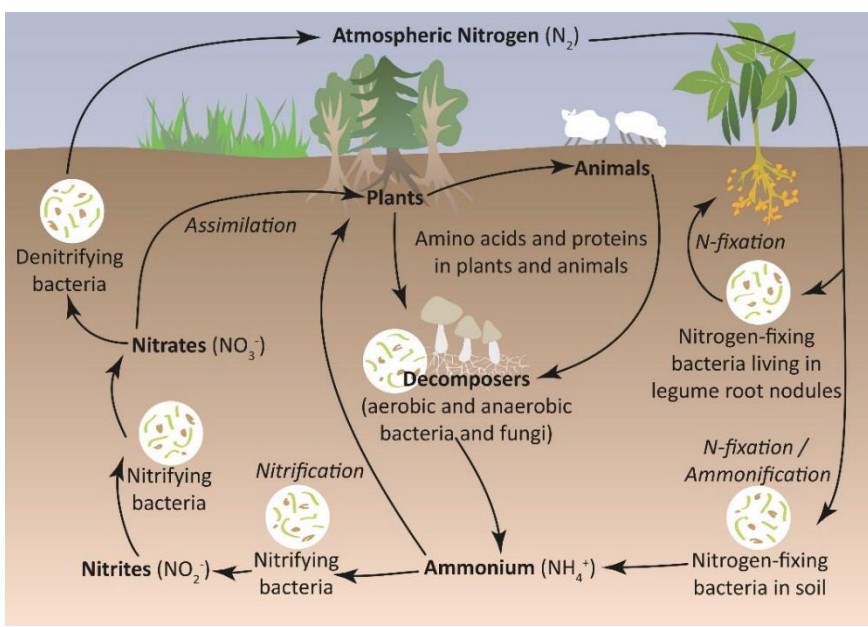


Figure 7: Schematic figure for terrestrial nitrogen cycle.

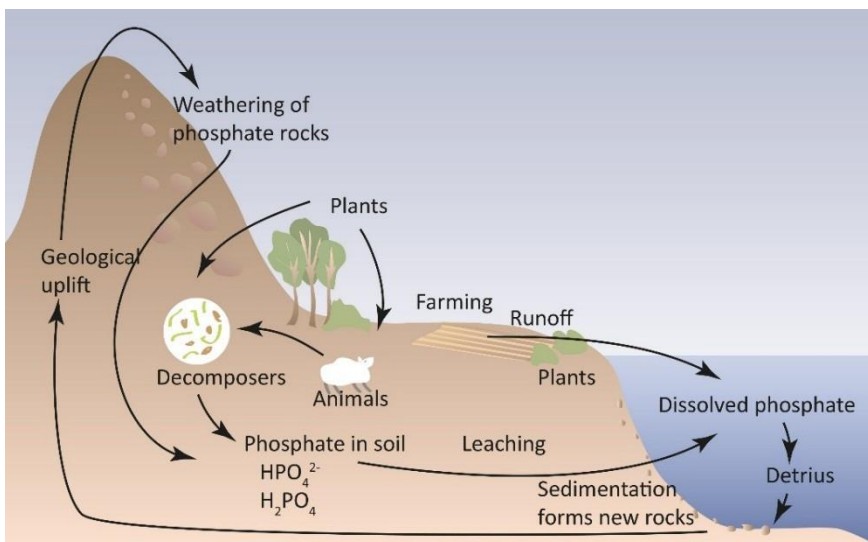


Figure 8: Schematic figure of the phosphorus cycle.


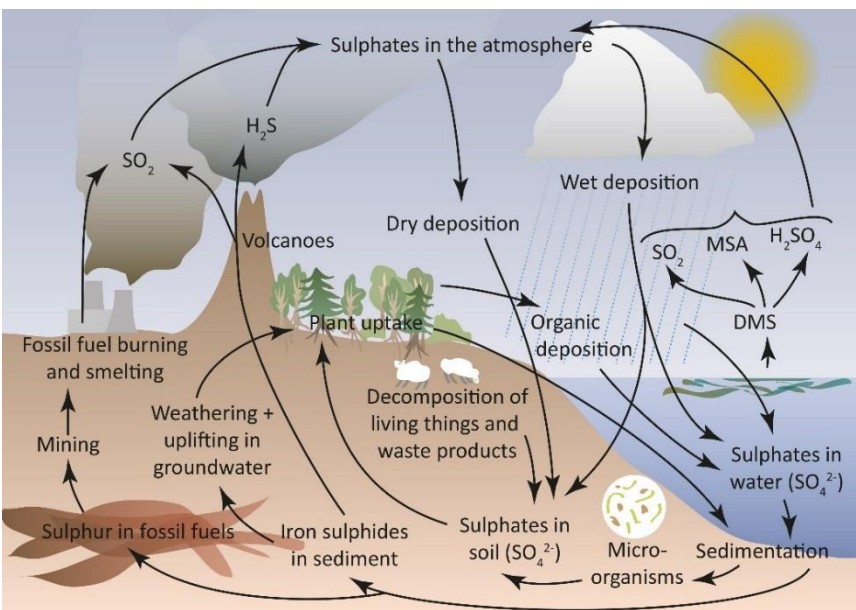


Figure 9: Schematic figure of the sulfur cycle.

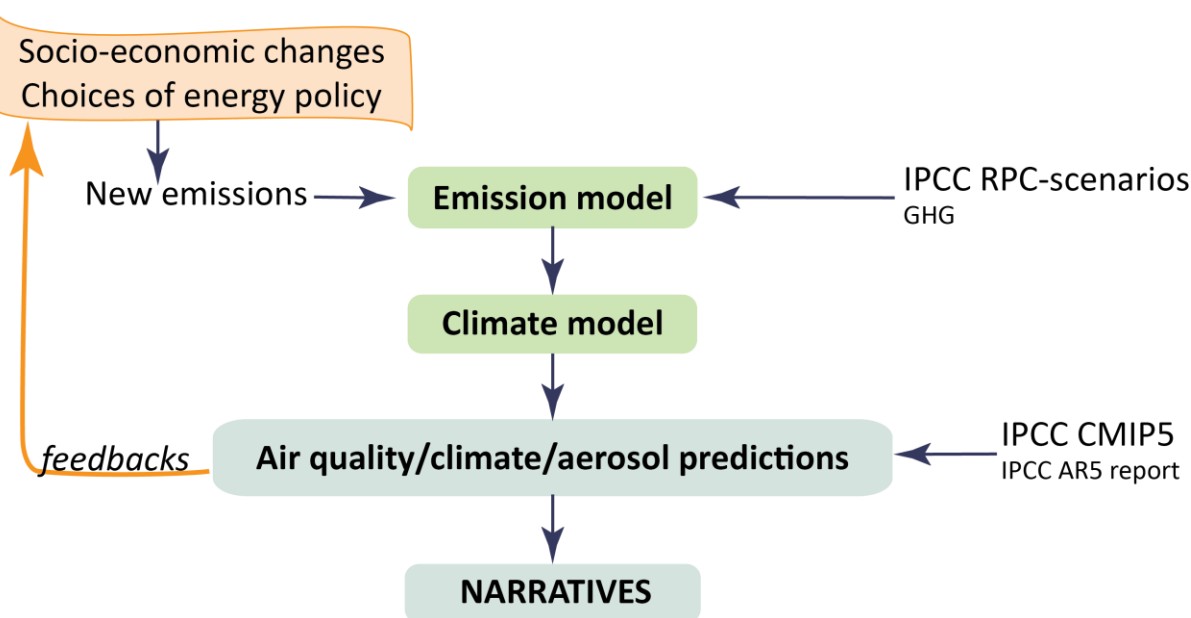



Figure 10. An example of the study approach to be implemented by PEEX for integrating
natural science and social science knowledge and generating climate predictions and narratives
of the Northern regions.