# Peer review of "Towards a holistic understanding of the feedbacks and interactions in the land - atmosphere - ocean- society continuum in the Northern Eurasian region"

_Atmospheric Chemistry and Physics, 2016_

## Referee Comment (RC1) · Anonymous Referee #1 · 12 May 2016

Recommendation

Dear editor and authors. Based on my analysis I recommend to accept the current paper subject to only minor revisions.

General comments

The paper aims at describing the mega-project "PEEX", and how the project leaders use an approach of research, infrastructures, societal impact and stakeholder com-

munication to address the environment issues in the northern Eurasian area in the atmosphere, soil and water ecosystems. But, the main goal is to review and identify these environmental issues. I think that this is invaluable information, which will be used as a reference for coming PEEX-papers and other authors using this paper as look-up reference for their own studies. Hence, this paper will be referenced multiple times. This paper is also an ethically correct and timely construction, since it can serve as an eye-opener for the public and for stakeholders and decision makers of the local or regional authorities, and can be used for mitigation or adaptation. Namely, this paper comprehensively summarizes the negative environmental impacts in the current geographical region, where there has been little motivation, attention or pressure on the authorities to act on the different environmental issues to date.

I base my review on my experience mainly related to the atmospheric environment, but am able to comment and understand also the other parts of the manuscript due to the easily understandable explanation provided by the authors. I think that the language does not suffer from any flaws, but is very easy to follow, and has a clear structure. I can see that the structure and language of the paper has been thought through with high rigor. The paper is written in a very general manner, without too detailed explanations of the environmental issues at hand. This is perhaps the largest drawback and strength of the paper at the same time. I reckon the readability of the paper would suffer very much if the text would be much longer. It is already long as it is. Hence, I think that the current length of the paper is a well justified compromise.

Specific comments

A number of misspellings in the article. Please revise accordingly. But, no sentence structure problems.

Introduction, line 100: Please replace "will" with "is". We are already facing issues with the grand challenges.

Introduction, lines 103-104. "These changes are also reflected from and linked with the

natural environments at large spatial scales." This sentence doesn't give added value to the paragraph. Please, consider removing.

Please create a table of contents, despite that this layout is not familiar to journal articles.

Chapter 2, lines 132-135. Maybe a little bit too much talking without concrete information. Consider removing.

Chapter 2, line 250. What does figure 2 have to do with the information in the paragraph? Please explain.

Chapter 2, lines 251-252. What does "turnover of soil carbon stocks" mean? Please explain.

Chapter 2, lines 275-284. Arctic "Browning" vs "greening". Which process is today dominating area-wise? On line 1168 you are stating that the greening dominates, but this is not clear here. Please clarify.

Figure 3. Very low picture resolution of map. Please improve.

Chapter 2. Lines 508-510. The stable atmospheric stratification, is that high pressure subsidence inversions, or other types? Please mention which type of large-scale stratification it is.

Chapter 2.2.3.2. How can the general electric circuit be used as diagnostic tool for climate studies. Please explain and give references.

Chapter 2. Line 625. "fall-winter energy loss from the ocean". What is this?

Chapter 2. Line 774-775 "The higher temperature response of aquatic ecosystems compared to terrestrial ecosystems. . .". How do you know that the temperature response is higher for aquatic ecosystems?

Chapter 2.4.3. "The impact of climate parameters, such as temperature (including sea-
* * *
Interactive
comment

sonal, weekly and daily gradients, and extreme values), strong winds, snowfall, snowstorms and precipitation should be investigated. Both the frequency and the duration of weather events should be considered. These climate parameters influence human health, tincidence of diseases,...". Not convincing that the climate change has major negative effects on human health in northern Eurasia. If your conclusion is that climate change is not affecting human health in a major negative way in northern Russia, please write this out.

Chapter 3.2. The acronyms NPP and HSR should be defined the first time these are mentioned.

Lines 1191-1194. Serious misspellings.

Chapter 3.4. Lines 1276-1283 is a repetition of the previous paragraph.

Chapter 4. Lines 1378-1393. What is it that you are really trying to communicate in this paragraph. It is so general, that it becomes very hard to understand. Please concretize.

Introduction: Please write at the end of the introduction chapter what is the outline of your report. In other words: describe shortly what is the goal of the different chapters in order for the reader to get a full picture of how the report is organized and how the different parts connect.

---

## Referee Comment (RC2) · Anonymous Referee #2 · 23 Aug 2016

Review for manuscript: Pan-Eurasian Experiment (PEEX): Towards holistic understanding of the feedbacks and interactions in the land–atmosphere–ocean–society continuum in the Northern Eurasian region" by Hanna K. Lappalainen et al. The article gives a comprehensive review of PEEX project in a very detailed way. Scientific issues of atmosphere and ecosystem due to environment change over Northern Eurasian regions and Arctic Ocean regions in the next decades are presented. Details are shown in separated sections, included the arctic-boreal natural environments how will play a

crucial role in the global climate; hydrological and carbon cycle; nitrogen, sulfur and phosphorus cycle. Especially, according to the industry andeconomy increase of Eastern Asia, there is a huge demand for natural resources, for example, natural gas, coal, petroleum and minerals. Those anthropogenic activities will emit vast pollutants to the atmosphere, river, lake and ocean, which will cause local and regional air pollution, climate change. The Northern Eurasian regions and Arctic Ocean region will play a crucial role in the global climate. So a novel research approach is needed actually, which not only identifies and tackles the relevant multi-disciplinary research questions, but is also able to make a holistic system analysis of the expected feedbacks. In this manuscript, the authors introduced the research agenda of the Pan- Eurasian Experiment (PEEX) successfully. The review article will be a holistic supplement for other published PEEX papers after addressing the following aspects. I suggest that the paper should be published in ACP after minor revise. Major comments: 1. Page 48 Line 1314, you mentioned that 'Sulfur emissions in China are rapidly increasing', please give some references about it. As I know, the emission of NO2 is increasing rapidly in China, while the increasing of SO2 is complex after 2008 Olympic Games. Minor comments: Page 2 Line 76 : 'Craduate University of Chinese Academy of Sciences' as 'Graduate University of Chinese Academy of Sciences' Figure 2&figure 3, with low resolution, are not very clear as other pictures, please update them. Page 6 line189: spelling mistake of 'atmosphere'

---

## Author Comment (AC1) · 20 Sep 2016

1. Introduction, line 100: replace "will" with "is" AUTHORS' RESPONSE: corrected according to referee comment

2. Introduction, line 103-104: consider removing the sentence AUTHORS' RE-SPONSE: the sentence has been removed.

3. Create "Table of contents" AUTHORS' RESPONSE: We have added the "Table of

contents" after the "Abstract" section

4. Chapter 2, lines 132-135; consider removing, no concrete information AUTHORS'
RESPONSE: we would like to keep this sentence.

5. Chapter 2, line 250. What does the figure 2 do with the information in this paragraph, explain' AUTHORS' RESPONSE: We have added the following sentence to clarify the link between text and Fig.2: "Liners trends in the annual maximum Normalized Difference Vegetation Index (NDVI) over 15 years in the Northern areas of the Yamalo-Nenets, Okrug region in Russia provides supporting evidence of the increasing biological activity and greening and potential to enhanced BVOC emissions (Fig.2). Furthermore, we have added the following to the figure 2. caption" Figure 2: Linear trends in the annual maximum Normalized Difference Vegetation Index (NDVI) obtained from analysis of the MODIS 0.25 km data product for 2000-2014 over the North-Western Siberia region in Russia. The trends are given in the NDVI changes per 15 years. The yellow colors show the decreasing NDVI, which corresponds to decreasing biological production; the blue colors show the increasing NDVI. More detailed analysis of the trends is given in Esau et al. (2016).

6. Chapter 2, lines 251-252. What does "turnover of soil carbon stocks" mean? Please explain AUTHORS' RESPONSE: We have removed to following text "and in the turnover of soil carbon stocks".

7. Chapter 2, lines 275-284. Arctic "Browning" vs. "greening. Which process is today dominating area-wise? On line 1168 you are stating that the greening dominates, but this is not clear here. Please clarify AUTHORS'RESPONSE: We have added the reference Phoenix, G.K., and J.W. Bjerke, J.W.:Arctic browning: extreme events and trends reversing arctic greening, Global Change Biology, 22, 2960–2962, 2016. And modified the lines 275-252 as following:

"However, browning as a proxy of decreased productivity has been observed during recent decades in many boreal regions (Lloyd and Bunn 2007), including vast territories

of Central Siberia together with a general downward trend in basal area increment after the mid-20th century (Berner et al., 2013) and the overall decline in greenness from 2011 to 2014 in Arctic regions (Phoenix and Bjerke 2016). Current predictions on the extent and magnitude of these processes vary significantly (Tchebakova et al., 2009; Hickler et al., 2012; Shvidenko et al., 2013). It has been estimated that the northward shift of bioclimatic zones in Siberia will be as large as 600 km by the end of this century (Tchebakova et al., 2009). By taking into account that the natural migration rate of boreal tree species cannot exceed 200-500 m per year, such a forecast implies major vegetation changes in huge areas. In addition, we need to have a deeper understanding on the future role of the browing process and re-analyze the model predictions of arctic greening; to what extent are they wrong, and why (Phoenix and Bjerke 2016)."

Figure 3. Very low resolution of map. Please improve AUTHORS' RESPONSE: we provide an improved version of the map 8. Chapter 2, lines 508-510. The stable atmospheric stratification, is that high pressure subsidence inversions, or other types? Please mention which type of large-scale stratification it is. AUTHORS' RESPONSE: Stable stratification is typical phenomena in the night time during summer. In Siberia, stable stratification takes place in the winter time and is independent on pressure . The text is modified accordingly.

There is no strong correlation between the atmospheric air stratification and type of air masses. These are different measures to characterize the atmosphere. Temperature inversions are formed in high pressure air masses in clear sky conditions, which enables the cooling of ground surface, commonly during nighttime and early morning. Stable atmospheric stratification can form also in other type of air masses, e.g. low pressure air masses with low winds or calm during nighttime. Atmospheric stability is characterized by Pasquill stability classes according to various meteorological parameters.

https://www.ready.noaa.gov/READYpgclass.php http://onlinelibrary.wiley.com/doi/10.1002/9780470935361.app1/pdf https://en.wikipedia.org/wiki/Outline_of_air_pollution_dispersion

9. Chapter 2.2.3.2, How can the general electric circuit be used as diagnostic tool for climate studies. Please explain and give references.ÂÍ AUTHROR' RESPONSE: We have added a reference Mareev E.: Global electric circuit research: achievements and prospects, Uspekhi Fizicheskikh Nauk and P N Lebedev Physics Institute of the Russian Academy of Sciences, Physics-Uspekhi, 53, 504- 511, DOI: 10.3367/UFNe.0180.201005h.052 and edited the text as following: Further exploration of the GEC to be part of the climate sytem studies, its effect on the balance between Earth ionosphere and global circuit, requires accurate modeling of the GEC stationary state and its dynamics (Mareev 2010).

10. Chapter 2, lines Line 774-775 "the higher temperature response of aquatic ecosystems compared to terrestrial. . ." How do you know that the temperature response is higher for aquatic ecosystem? AUTHORS' RESPONSE: Yvon-Durocher et al., 2012 have used large dataset respiratory measurements demonstrating and showed "show that the sensitivity of ecosystem respiration to seasonal changes in temperature is remarkably similar for diverse environments encompassing lakes, rivers, estuaries, the open ocean and forested and non-forested terrestrial ecosystems, with an average activation energy similar to that of the respiratory complex3. By contrast, annual ecosystem respiration shows a substantially greater temperature dependence across aquatic versus terrestrial ecosystems that span broad geographic gradients in temperature.". We have added "observed": The observed higher temperature response of aquatic ecosystems

11. Chapter 2.4.3 "The impact of climate parameters, such as temperature (including sea seasonal, weekly), strong winds, snowfall, snowstorms and precipitation should be investigated. Both the frequency and the duration of weather extremes. . .incidence of diseases". Not convincing that the climate change major negative effect on human health in northern Eurasia. If your conclusion is that climate change is not affecting human health in a major negative way in northern Russia. Please write this out. AUTHORS' RESPONSE: We modified the text as following: the living conditions mostly in

Eastern part of the Northern Eurasian societies,

12. Chapter 3.2. The acronyms NPP and HSR should be defined the first time these are mentioned. AUTHORS' RESPONSE: we have defined the acronyms: terrestrial net primary production (NPP) and heterotrophic soil respiration (HSR)

13. Lines 1191-1194. Serious misspellings' AUTHORS' RESPONSE:We have corrected the text as following: At the same time the fluvial export by the largest rivers considered to be an order of magnitude less than coastal erosion in the East Siberian Arctic Shelf (Semiletov et al, 2011). The Lena's particulate organic carbon export is two orders of magnitude less than the annual input of eroded terrestrial carbon onto the shelf of the Laptev and East Siberian seas.

14. Chapter 3.4 Lines 1276-1283 is a repretion of the previous paragraph AUTHORS' RESPONSE: To avoid a repretion we have removed the sentence: "Southwestern Siberian soils have lately been reported to contain high concentrations of plant-available phosphorus (Achant et al., 2013), which may enhance carbon sequestration of the ecosystems, if nitrogen is not too limited."

15. Chapter 4 Lines 1378-1393. What is it you are really trying to communicate in this paragraph. It is general, that it becomes very hard to understand. Please concretize AUTHORS' RESPONSE: We have shorten and edited the text as follows:

"PEEX is interested in developing methodologies for integrating natural science and social science knowledge as part of the operational Earth sustainable system questions (Schellnhuber et al. 2004). The first-priority tasks in this case is to establish an integrated geographical information background (Ribeiro et al., 2009; Hunt and Sanchez-Rodriquez, 2009; Shvidenko et al. 2010; Skryzhevska et al., 2015). A common information background would be the first step serving the development of a common language of integrated studies. For example, we need spatially and temporally explicit descriptions of terrestrial ecosystems, landscapes, atmosphere and hydrosphere. A common information background would be a unified base for the PEEX modelling platform and for the development of integrated modelling clusters, which could combine ecological, economic and social dimensions. It could provide a historical background for the future trajectories of land cover, state and resilience of ecosystems, stability of landscapes, and dynamics of environmental indicators of environment. The already exiting Integrated Land Information System could be utilized here for combining all historical knowledge about the region and all scientific results obtained by past, current and future studies across the region (e.g. Schepaschenko et al., 2011; Shvidenko and Schepaschenko, 2014)."

16. Introduction: please write at the end of the introduction chapter what is the outline of your report. In other words_ describe shortly what is the goal of different chapters in order for the reader to get a full picture of how the report is organized and how the different parts connect. AUTHORS' RESPONSE: To clarify the structure of the paper we have added the "Table of Content" before the "Introduction". The structure and goals of the different chapters is introduced in the three last paragraphs of the Section "2.System perspective approach". We think that now having the "Table of Content", as proposed by the both referees, clearly provides the full picture of the report, it's goals are, connections between different parts. The introduction of the report fits better to the end of chapter-2 than in the end of "Introduction". Having it in Chapter 2. we are addressing the "system based" orientation of the approach.

Please also note the supplement to this comment:
http://www.atmos-chem-phys-discuss.net/acp-2016-186/acp-2016-186-AC1-supplement.pdf

---

## Author Comment (AC2) · 20 Sep 2016

ANNEX-1 Anonymous Referee 2

Anonymous Referee 2 1. Page 34, line 1314; you mention that "Sulfur emissions in China are rapidly increasing". Please give some references about it. As I know, the emissions of NO2 is increasing rapidly in China, while the increasing of SO2 is complex after 2008 Olympic Games. AUTHORS' RESPONSE: We refer to Lu & Zhang:

Sulfur dioxide and primary carbonaceous aerosol emissions in China and India, Atmos. Chem. Phys., 11, 9839-9864, 2011 where they say that "SO2 emissions first increased by 61 % to 34.0 Tg in 2006, and then decreased by 9.2 % to 30.8 Tg in 2010 due to the wide application of flue-gas desulfurization (FGD) equipment in power plants.". The text has been modified as following: " "For example, sulfur emissions in China creased rapidly until 2006, and then decreased by 9.2 % to 30.8 Tg in 2010 due to the wide application of flue-gas desulfurization (FGD) equipment in power plants (Lu and Zhang 2011), while emissions in Europe have significantly decreased during the last decades."

2. Page 2 line 76 "Craduate University of Chinese Academy of Sciences" as "Graduate University of Chinese Academy of Sciences" AUTHORS' RESPONSE: corrected

3. Fig 2 & Fig 3 with low resolution are not clear as other picture, please update them. AUTHORS' RESPONSE: new higher resolution figures are provided.

4. Page 6 line 189: spelling mistake of "atmosphere AUTHORS' RESPONSE: corrected

Please also note the supplement to this comment:
http://www.atmos-chem-phys-discuss.net/acp-2016-186/acp-2016-186-AC2-supplement.pdf

[Figure]

**Fig. 1.**

[Figure]

**Fig. 2.**

[Figure]

**Fig. 3.**

[Figure]

**Fig. 4.**

[Figure]

**Fig. 5.**

[Figure]

**Fig. 6.**

[Figure]

**Fig. 7.**

[Figure]

**Fig. 8.**

[Figure]

**Fig. 9.**

[Figure]

**Fig. 10.**

**Supplement:**

[Figure]

Helsinki 09.Sep.2016

Ref.       Authors' response to Referee comments on the ACPD manuscript titled "*Pan-Eurasian Experiment (PEEX): Towards holistic understanding of the feedbacks and interactions in the land - atmosphere - ocean- society continuum in the Northern Eurasian region*"

Dear Editor,

Thank you for the possibility to revise and submit our ACPD manuscript Lappalainen et al. *"Pan-Eurasian Experiment (PEEX): Towards holistic understanding of the feedbacks and interactions in the land - atmosphere - ocean- society continuum in the Northern Eurasian region*" for Journal of Atmospheric Chemistry and Physics - PEEX Special issue. We also thank the two anonymous reviewers for constructive comments on our manuscript. In the following, we have responded to the associate editors and the reviewers' comments point-by-point (Annex 1, 2), and the corresponding changes has been applied to the manuscript (indicated by red font). We have also included some other corrections or improvements, which are introduced and listed after referee comments in Annex 3. We hope that the revised manuscript version could be considered for publication in the J.ACP.

Sincerely,

Hanna Lappalainen on behalf of all authors
University of Helsinki / Finnish Meteorological Institute

cc: Veli-Matti Kerminen
University of Helsinki

e-mail:      hanna.k.lappalainen(at)helsinki.fi
tel.        +358 050 434 1710

Fysiikan laitos, PL 64 (Gustaf Hällströmin katu 2), 00014 Helsingin yliopisto
Puhelin 02941 50600, faksi 02941 50610, www.helsinki.fi
Institutionen för fysik, PB 64 (Gustaf Hällströms gata 2), FI-00014 Helsingfors universitet
Telefon +358 2941 50600, fax +358 2941 50610, www.helsinki.fi
Department of Physics, P.O. Box 64 (Gustaf Hällströmin katu 2), FI-00014 University of Helsinki
Telephone +358 2941 50600, fax +358 2941 50610, www.helsinki.fi

HELSINGIN YLIOPISTO
HELSINGFORS UNIVERSITET
UNIVERSITY OF HELSINKI

MATEMAATTIS-LUONNONTIETEELLINEN TIEDEKUNTA
MATEMATISK-NATURVETENSKAPLIGA FAKULTETEN
FACULTY OF SCIENCE

[Figure]

**ANNEX-1    Anonymous  Referee 1**

1.  Introduction, line 100: replace "will" with "is"
    AUTHORS' RESPONSE: corrected according  to referee comment

2.  Introduction, line 103-104: consider removing the sentence
    AUTHORS' RESPONSE:  the sentence has been removed.

3.  Create "Table of contents"
    AUTHORS' RESPONSE: We have added the "Table of contents" after the "Abstract" section

4.  Chapter 2, lines 132-135; consider removing, no concrete information
    AUTHORS' RESPONSE: we would like to keep this sentence.

5.  Chapter 2, line 250. What does the figure 2 do with the information in this paragraph, explain'
    AUTHORS' RESPONSE: We have added the following sentence to clarify the link between text and Fig.2: "Liners trends in the annual maximum Normalized Difference Vegetation Index (NDVI) over 15 years in the Northern areas of the Yamalo-Nenets, Okrug region in Russia  provides supporting evidence of the increasing  biological activity and greening   and   potential to enhanced BVOC emissions (Fig.2). Furthermore, we have added the following to the figure 2. caption" Figure 2: Linear trends in the annual maximum Normalized Difference Vegetation Index (NDVI) obtained from analysis of the MODIS 0.25 km data product for 2000-2014 over the North-Western Siberia region in Russia. The trends are given in the NDVI changes per 15 years. The yellow colors show the decreasing NDVI, which corresponds to decreasing biological production; the blue colors show the increasing NDVI. More detailed analysis of the trends is given in Esau et al. (2016).

6.  Chapter 2, lines 251-252. What does "turnover of soil carbon stocks" mean? Please explain
    AUTHORS' RESPONSE: We have removed to following text "and in the turnover of soil carbon stocks".

7.  Chapter 2, lines 275-284. Arctic "Browning" vs. "greening. Which process is today dominating area-wise? On line 1168 you are stating that the greening dominates, but this is not clear here. Please clarify
    AUTHORS'RESPONSE:   We have added the reference Phoenix, G.K., and J.W. Bjerke, J.W.:Arctic browning: extreme events and trends reversing arctic greening, Global Change Biology, 22, 2960–2962, 2016. And modified the lines 275-252 as following:

    "However, browning as a proxy of decreased productivity has been observed during recent decades in many boreal regions (Lloyd and Bunn 2007), including vast territories of Central Siberia together with a general downward trend in basal area increment after the mid-20th century (Berner et al., 2013) and the overall decline in greenness from 2011 to 2014 in Arctic regions  (Phoenix and Bjerke 2016). Current predictions on the extent and magnitude of these processes vary significantly (Tchebakova et al., 2009; Hickler et al., 2012; Shvidenko et al., 2013). It has been estimated that the northward shift of bioclimatic zones in Siberia will be as large as 600 km by the end of this century (Tchebakova et al., 2009). By taking into account that the natural migration rate of boreal tree species cannot exceed 200-500 m per year, such a forecast implies major vegetation changes in huge areas. In addition, we need to have a deeper understanding on the future role of the browing process and re-analyze the model predictions of arctic greening; to what extent are they wrong, and why (Phoenix and Bjerke 2016)."

Fysiikan laitos, PL 64 (Gustaf Hällströmin katu 2), 00014 Helsingin yliopisto
Puhelin 02941 50600, faksi 02941 50610, www.helsinki.fi
Institutionen för fysik, PB 64 (Gustaf Hällströms gata 2), FI-00014 Helsingfors universitet
Telefon +358  2941 50600, fax +358 2941 50610, www.helsinki.fi
Department of Physics, P.O. Box 64 (Gustaf Hällströmin katu 2), FI-00014 University of Helsinki
Telephone +358 2941 50600, fax +358 2941 50610, www.helsinki.fi

HELSINGIN YLIOPISTO
HELSINGFORS UNIVERSITET
UNIVERSITY OF HELSINKI

MATEMAATTIS-LUONNONTIETEELLINEN TIEDEKUNTA
MATEMATISK-NATURVETENSKAPLIGA FAKULTETEN
FACULTY OF SCIENCE

[Figure]

Figure 3. Very low resolution of map. Please improve

AUTHORS' RESPONSE: we provide an improved version of the map

8. Chapter 2, lines 508-510. The stable atmospheric stratification, is that high pressure subsidence inversions, or other types? Please mention which type of large-scale stratification it is.

   AUTHORS' RESPONSE: Stable stratification is typical phenomena in the night time during summer. In Siberia, stable stratification takes place in the winter time and is independent on pressure . The text is modified accordingly.

   There is no strong correlation between the atmospheric air stratification and type of air masses. These are different measures to characterize the atmosphere. Temperature inversions are formed in high pressure air masses in clear  sky conditions, which enables the cooling of ground surface, commonly during nighttime and early morning. Stable atmospheric stratification can form also in other type of air masses, e.g. low pressure air masses with low winds or calm during nighttime. Atmospheric stability is characterized by Pasquill stability classes according to various meteorological parameters.

   https://www.ready.noaa.gov/READYpgclass.php
   http://onlinelibrary.wiley.com/doi/10.1002/9780470935361.app1/pdf
   https://en.wikipedia.org/wiki/Outline_of_air_pollution_dispersion

9. Chapter 2.2.3.2, How can the general electric circuit be used as diagnostic tool for climate studies. Please explain and give references.¨

   AUTHROR' RESPONSE: We have added a reference Mareev E.: Global electric circuit research: achievements and prospects, Uspekhi Fizicheskikh Nauk and P N Lebedev Physics Institute of the Russian Academy of Sciences, Physics-Uspekhi, 53, 504- 511, DOI: 10.3367/UFNe.0180.201005h.052 and edited the text as following: Further exploration of the GEC to be part of the climate sytem studies, its effect on the balance between  Earth ionosphere and global circuit, requires accurate modeling of the GEC stationary state and its dynamics (Mareev 2010).

10. Chapter 2, lines Line 774-775 "the higher temperature response of aquatic ecosystems compared to terrestrial…" How do you know that the temperature response is higher for aquatic ecosystem?

    AUTHORS' RESPONSE: Yvon-Durocher et al., 2012 have used large dataset respiratory measurements demonstrating and showed "show that the sensitivity of ecosystem respiration to seasonal changes in temperature is remarkably similar for diverse environments encompassing lakes, rivers, estuaries, the open ocean and forested and non-forested terrestrial ecosystems, with an average activation energy similar to that of the respiratory complex[3]. By contrast, annual ecosystem respiration shows a substantially greater temperature dependence across aquatic versus terrestrial ecosystems  that span broad geographic gradients in temperature.". We have added "observed": The observed higher temperature response of aquatic ecosystems

11. Chapter 2.4.3 "The impact of climate parameters, such as temperature (including sea seasonal, weekly), strong winds, snowfall, snowstorms and precipitation should be investigated. Both the frequency and the duration of weather extremes…incidence of diseases". Not convincing that the climate change major negative effect on human health in northern Eurasia. If your

Fysiikan laitos, PL 64 (Gustaf Hällströmin katu 2), 00014 Helsingin yliopisto
Puhelin 02941 50600, faksi 02941 50610, www.helsinki.fi
Institutionen för fysik, PB 64 (Gustaf Hällströms gata 2), FI-00014 Helsingfors universitet
Telefon +358  2941 50600, fax +358 2941 50610, www.helsinki.fi
Department of Physics, P.O. Box 64 (Gustaf Hällströmin katu 2), FI-00014 University of Helsinki
Telephone +358 2941 50600, fax +358 2941 50610, www.helsinki.fi

HELSINGIN YLIOPISTO
HELSINGFORS UNIVERSITET
UNIVERSITY OF HELSINKI

MATEMAATTIS-LUONNONTIETEELLINEN TIEDEKUNTA
MATEMATISK-NATURVETENSKAPLIGA FAKULTETEN
FACULTY OF SCIENCE

conclusion is that climate change is not affecting human health in a major negative way in northern Russia. Please write this out.
AUTHORS' RESPONSE: We modified the text as following: the living conditions mostly in Eastern part of the Northern Eurasian societies,

12. Chapter 3.2. The acronyms NPP and HSR should be defined the first time these are mentioned.
AUTHORS' RESPONSE: we have defined the acronyms: terrestrial net primary production (NPP) and heterotrophic soil respiration (HSR)

13. Lines 1191-1194. Serious misspellings´
AUTHORS' RESPONSE:We have corrected the text as following: At the same time the fluvial export by the largest rivers considered to be an order of magnitude less than coastal erosion in the East Siberian Arctic Shelf (Semiletov et al, 2011). The Lena's particulate organic carbon export is two orders of magnitude less than the annual input of eroded terrestrial carbon onto the shelf of the Laptev and East Siberian seas.

14. Chapter 3.4 Lines 1276-1283 is a repretion of the previous paragraph
AUTHORS' RESPONSE: To avoid a repretion we have removed the sentence: "Southwestern Siberian soils have lately been reported to contain high concentrations of plant-available phosphorus (Achant et al., 2013), which may enhance carbon sequestration of the ecosystems, if nitrogen is not too limited."

15. Chapter 4 Lines 1378-1393. What is it you are really trying to communicate in this paragraph. It is general, that it becomes very hard to understand. Please concretize
AUTHORS' RESPONSE: We have shorten and edited the text as follows:

"PEEX is interested in developing methodologies for integrating natural science and social science knowledge as part of the operational Earth sustainable system questions (Schellnhuber et al. 2004). The first-priority tasks in this case is to establish an integrated geographical information background (Ribeiro et al., 2009; Hunt and Sanchez-Rodriquez, 2009; Shvidenko et al. 2010; Skryzhevska et al., 2015). A common information background would be the first step serving the development of a common language of integrated studies. For example, we need spatially and temporally explicit descriptions of terrestrial ecosystems, landscapes, atmosphere and hydrosphere. A common information background would be a unified base for the PEEX modelling platform and for the development of integrated modelling clusters, which could combine ecological, economic and social dimensions. It could provide a historical background for the future trajectories of land cover, state and resilience of ecosystems, stability of landscapes, and dynamics of environmental indicators of environment. The already exiting Integrated Land Information System could be utilized here for combining all historical knowledge about the region and all scientific results obtained by past, current and future studies across the region (e.g. Schepaschenko et al., 2011; Shvidenko and Schepaschenko, 2014)."

16. Introduction: please write at the end of the introduction chapter what is the outline of your report. In other words_ describe shortly what is the goal of different chapters in order for the reader to get a full picture of how the report is organized and how the different parts connect.
AUTHORS' RESPONSE: To clarify the structure of the paper we have added the "Table of Content" before the "Introduction". The structure and goals of the different chapters is introduced in the three last paragraphs of the Section "2.System perspective approach". We think that now having the "Table of Content", as proposed by the both referees, clearly

Fysiikan laitos, PL 64 (Gustaf Hällströmin katu 2), 00014 Helsingin yliopisto
Puhelin 02941 50600, faksi 02941 50610, www.helsinki.fi
Institutionen för fysik, PB 64 (Gustaf Hällströms gata 2), FI-00014 Helsingfors universitet
Telefon +358 2941 50600, fax +358 2941 50610, www.helsinki.fi
Department of Physics, P.O. Box 64 (Gustaf Hällströmin katu 2), FI-00014 University of Helsinki
Telephone +358 2941 50600, fax +358 2941 50610, www.helsinki.fi

HELSINGIN YLIOPISTO
HELSINGFORS UNIVERSITET
UNIVERSITY OF HELSINKI

MATEMAATTIS-LUONNONTIETEELLINEN TIEDEKUNTA
MATEMATISK-NATURVETENSKAPLIGA FAKULTETEN
FACULTY OF SCIENCE

provides the full picture of the report, it's goals are, connections between different parts. The introduction of the report fits better to the end of chapter-2 than in the end of "Introduction". Having it in Chapter 2. we are addressing the "system based" orientation of the approach.

HELSINGIN YLIOPISTO
HELSINGFORS UNIVERSITET
UNIVERSITY OF HELSINKI

MATEMAATTIS-LUONNONTIETEELLINEN TIEDEKUNTA
MATEMATISK-NATURVETENSKAPLIGA FAKULTETEN
FACULTY OF SCIENCE

Fysiikan laitos, PL 64 (Gustaf Hällströmin katu 2), 00014 Helsingin yliopisto
Puhelin 02941 50600, faksi 02941 50610, www.helsinki.fi
Institutionen för fysik, PB 64 (Gustaf Hällströms gata 2), FI-00014 Helsingfors universitet
Telefon +358  2941 50600, fax +358 2941 50610, www.helsinki.fi
Department of Physics, P.O. Box 64 (Gustaf Hällströmin katu 2), FI-00014 University of Helsinki
Telephone +358 2941 50600, fax +358 2941 50610, www.helsinki.fi

[Figure]

**ANNEX-1   Anonymous Referee 2**

**Anonymous Referee 2**

1. Page 34, line 1314; you mention that "Sulfur emissions in China are rapidly increasing". Please give some references about it. As I know, the emissions of NO2 is increasing rapidly in China, while the increasing of SO2 is complex after 2008 Olympic Games.
   AUTHORS' RESPONSE:  We refer to Lu &  Zhang: Sulfur dioxide and primary carbonaceous aerosol emissions in China and India, Atmos. Chem. Phys., 11, 9839-9864, 2011 where they say that "$SO_2$ emissions first increased by 61 % to 34.0 Tg in 2006, and then decreased by 9.2 % to 30.8 Tg in 2010 due to the wide application of flue-gas desulfurization (FGD) equipment in power plants.". The text has been modified as following: "
   "For example, sulfur emissions in China creased rapidly until 2006, and then decreased by 9.2 % to 30.8 Tg in 2010 due to the wide application of flue-gas desulfurization (FGD) equipment in power plants (Lu and Zhang 2011), while emissions in Europe have significantly decreased during the last decades."

2. Page 2 line 76 "Craduate University of Chinese Academy of Sciences" as "Graduate University of Chinese Academy of Sciences"
   AUTHORS' RESPONSE: corrected

3. Fig 2 & Fig 3 with low resolution are not clear as other picture, please update them.
   AUTHORS' RESPONSE: new higher resolution figures are provided.

4. Page 6 line 189: spelling mistake of "atmosphere
   AUTHORS' RESPONSE: corrected

**ANNEX 3   Other corrections**

- We have re-made the language checking.
- We have added Prof. Meinrat O. Andreae, MaxPlanck Institute, in our co-.author list
- Last name of Ella-Maria Kyrö is changed to Ella-Maria Duplissy

**Fysiikan laitos, PL 64 (Gustaf Hällströmin katu 2), 00014 Helsingin yliopisto**
**Puhelin 02941 50600, faksi 02941 50610, www.helsinki.fi**
**Institutionen för fysik, PB 64 (Gustaf Hällströms gata 2), FI-00014 Helsingfors universitet**
**Telefon +358  2941 50600, fax +358 2941 50610, www.helsinki.fi**
**Department of Physics, P.O. Box 64 (Gustaf Hällströmin katu 2), FI-00014 University of Helsinki**
**Telephone +358 2941 50600, fax +358 2941 50610, www.helsinki.fi**

HELSINGIN YLIOPISTO
HELSINGFORS UNIVERSITET
UNIVERSITY OF HELSINKI

MATEMAATTIS-LUONNONTIETEELLINEN TIEDEKUNTA
MATEMATISK-NATURVETENSKAPLIGA FAKULTETEN
FACULTY OF SCIENCE